# Inseparable RNA binding and chromatin modification activities of a nucleosome-interacting surface in EZH2

Emma H. Gail[1,5], Evan Healy[1,5], Sarena F. Flanigan[1], Natasha Jones[1], Xiao Han Ng [1], Michael Uckelmann[1], Vitalina Levina[1], Qi Zhang [1,2,3] ✉ & Chen Davidovich [1,4] ✉

Polycomb repressive complex 2 (PRC2) interacts with RNA in cells, but there is no consensus on how RNA regulates PRC2 canonical functions, including chromatin modification and the maintenance of transcription programs in lineage-committed cells. We assayed two separation-of-function mutants of the PRC2 catalytic subunit EZH2, defective in RNA binding but functional in methyltransferase activity. We find that part of the RNA-binding surface of EZH2 is required for chromatin modification, yet this activity is independent of RNA. Mechanistically, the RNA-binding surface within EZH2 is required for chromatin modification in vitro and in cells, through interactions with nucleosomal DNA. Contrarily, an RNA-binding-defective mutant exhibited normal chromatin modification activity in vitro and in lineage-committed cells, accompanied by normal gene repression activity. Collectively, we show that part of the RNA-binding surface of EZH2, rather than the RNA-binding activity per se, is required for the histone methylation in vitro and in cells, through interactions with the substrate nucleosome.

RNA is found in most compartments of the nucleus and is required for their three-dimensional (3D) structural organization[1]. RNA binds to various chromatin modifiers and is proposed to regulate them, although the molecular mechanisms and their functional consequences are often not under consensus (reviewed in ref. 2). One chromatin modifier that has been extensively studied in the context of RNA-mediated regulation is the repressive histone methyltransferase PRC2 (ref. 3). PRC2 is required for normal development and for the maintenance of transcription programs in lineage-committed cells[4]. At the molecular level, PRC2 introduces the trimethyl modification to lysine 27 of histone H3 (H3K27me3), which is required for maintaining genes in a repressed state[5]. PRC2 interacts with thousands of transcripts in cells with rather broad specificity[6–8]. The functional consequences of

PRC2–RNA interactions are the subject of an ongoing debate (reviewed in refs. 3,9). On one hand, RNA inhibits the histone methyltransferase activity of PRC2 (refs. 10–12), competes against nucleosomes for binding to PRC2 (refs. 13,14) and removes PRC2 from genes[15]. On the other hand, RNA was previously proposed to recruit PRC2 to chromatin[16] and more recently reported to be essential for PRC2 chromatin occupancy[17] and polycomb-mediated genome architecture[18]. Hence, at this time there is no clarity or consensus on how RNA regulates the key canonical functions of PRC2: introducing the H3K27me3 mark to chromatin and enforcing lineage commitment through maintaining transcription programs[4].

An inherent challenge in determining how RNA regulates PRC2 was the difficulty in separating the RNA binding from other biochemical

[1]Department of Biochemistry and Molecular Biology, Biomedicine Discovery Institute, Faculty of Medicine, Nursing and Health Sciences, Monash University, Clayton, Victoria, Australia. [2]South Australian immunoGENomics Cancer Institute (SAiGENCI), Faculty of Health and Medical Sciences, University of Adelaide, Adelaide, South Australia, Australia. [3]EMBL-Australia at SAiGENCI, Adelaide, South Australia, Australia. [4]EMBL-Australia, Clayton, Victoria, Australia. [5]These authors contributed equally: Emma H. Gail, Evan Healy. ✉e-mail: qi.z.zhang@monash.edu; chen.davidovich@monash.edu

functions of PRC2. Separation-of-function mutations perturb one biochemical function without affecting others, allowing dissection of the contribution of specific biochemical properties of a given protein. Separation-of-function mutations in PRC2, which reduce its affinity for RNA without affecting its methyltransferase activity, have already been identified on the basis of in vitro studies[19,20]. One of these separation-of-function mutations, which includes ten substitutions in two separate patches in the catalytic subunit EZH2 (ref. 19), has already been tested in human pluripotent stem cells in two recent studies[17,18]. This mutant—termed mt2 herein (Fig. 1a)—exhibited aberrant targeting of PRC2 to chromatin, impaired deposition of the H3K27me3 mark[17], loss of PRC2-associated DNA looping[18] and cardiomyocyte differentiation defects[17]. These data were used to support a model where PRC2 requires its RNA-binding activity for its chromatin localization and function[17].

PRC2 uses multiple faces to contact chromatin. For instance, PRC2 can simultaneously contact the nucleosomal DNA of two adjacent nucleosomes[21] and also interact with the DNA linker that resides between nucleosomes[14]. This is conceptually similar to the case of RNA binding, which involves multiple dispersed patches in PRC2 (refs. 19,20,22). Hence, we reasoned that some PRC2 mutants might be bona fide separation-of-function mutants—defective in RNA binding and active in the methyltransferase of chromatin—while other mutants might be defective in the methylation of only certain substrates. We anticipated that comparing such different RNA-binding-defective mutants would allow us to determine the molecular function of different RNA-binding sites in PRC2.

Herein, we examine two separation-of-function mutations in EZH2 that are defective in RNA binding but catalytically active against isolated H3 histones. We directly compared these mutants against a catalytically defective EZH2 mutant, which serves as a background for methyltransferase loss-of-function. In vitro, we show that one of the RNA-binding-defective mutants (mt1; Fig. 1a) is a bona fide separation-of-function mutant, active in the methylation of H3 histones, nucleosomes and chromatin. The other mutant (mt2; Fig. 1a), which was considered a separation-of-function mutant in earlier studies in vitro[17,19] and in cells[17,18], is in fact defective in the modification of H3 histones while they are in the context of chromatin. We then compared these three mutants in cells, aiming to dissect the RNA-binding activity from the catalytic activity of PRC2. Our data indicate that a portion of the RNA-binding surface of PRC2 is as important as its catalytic activity for the canonical functions of PRC2 in cells: histone modification, gene repression and the maintenance of cell identity. Yet, these molecular functions of PRC2 were not substantially affected by a separation-of-function mutation in EZH2, defective in RNA binding and active in chromatin modification. Collectively, our data indicate that part of the RNA-binding surface of PRC2, but not its RNA-binding activity per se, is required for the modification of chromatin by PRC2.

## Results

### A portion of the RNA-binding surface of EZH2 is dispensable for the methylation of histone tails but required for the modification of chromatin in vitro

To identify functions of the RNA-binding surfaces of PRC2, we set out to assay two RNA-binding-defective separation-of-function EZH2 mutants that were identified in previous studies[19,20]. These EZH2 mutants, mt1 (ref. 20) and mt2 (ref. 19), include three and ten nonoverlapped amino acids, respectively (Fig. 1a), and are both catalytically active but defective in RNA binding. The majority of amino acids mutated in mt1 and mt2 are evolutionarily conserved (Supplementary Fig. 1a). We expressed and purified the four-subunit human PRC2 core complex, including EED, SUZ12, RBBP4 and either the wild-type or mutant EZH2 (Fig. 1c and Supplementary Fig. 1b,c). Quantitative RNA-binding assays confirmed that both PRC2 mutants mt1 and mt2 are defective in RNA binding, although the defect is more obvious in the case of mt1 (Supplementary Fig. 1d–h), which qualitatively in agreement with previous studies[19,20].

Nucleosome-binding assays, using electrophoretic mobility shift assay (EMSA), indicated that both mt1 and mt2 bind to nucleosomes with a similar affinity to the wild type (Supplementary Fig. 1i). This observation persisted whether the nucleosome core particle (NCP) probes included a linker DNA (NCP$_{182}$) or not (NCP$_{147}$). Yet, we noticed that a few of the amino acids that were mutated in mt2 reside in close proximity to the DNA of the substrate nucleosome (marked in red in Fig. 1b). This led us to hypothesize that mt2 might not be able to interact with chromatin in a conformation required for the efficient modification of histone tails. To test this hypothesis, we assayed the methyltransferase activity of mt1 and mt2 against three types of substrates that were previously used to characterize mt2 (ref. 17): isolated H3 histones, mononucleosomes and chromatin. We used unmodified human histone proteins in all the substrates. Unmodified (naïve) chromatin was reconstituted using unmodified human histones and a PCR-amplified 3.6-kilobase pair (kb) DNA from the human *ATOH1* locus (Supplementary Fig. 1j,k), which is repressed by PRC2 in most lineages[23]. The wild-type PRC2 and mt1 exhibited similar activities against H3 histones (Fig. 1d, left, and Supplementary Fig. 2a), mononucleosomes (Supplementary Fig. 2b) and naïve chromatin (Fig. 1d, right, and Supplementary Fig. 2c). This indicates that mt1 is a separation-of-function mutant, defective in RNA binding but active in the methylation of H3 histones within the context of nucleosomes or chromatin.

The histone methyltransferase activity of mt2 resembled that of the wild-type PRC2 when reacted against H3 histones externally to the context of chromatin (Fig. 1d, left, and Supplementary Fig. 2a), in agreement with previous studies[17,19]. Yet, mt2 was defective in modifying histones while they are in the context of nucleosomes (Supplementary Fig. 2b) or chromatin (Fig. 1d, right, and Supplementary Fig. 2c). These results indicate that mt2 separates the catalytic activity of PRC2 (Fig. 1d, left, and Supplementary Fig. 2a) from its chromatin modification activity (Fig. 1d, right, and Supplementary Fig. 2c). This also indicates

**Fig. 1 | An RNA-binding surface in EZH2 is required for the methylation of naïve chromatin in an RNA-independent manner. a**, EZH2 mutants used in this study. In parentheses are citations for studies where a given activity was assayed, unless not determined (ND). **b**, Amino acids that mutated in this study are indicated (in blue, red and orange for mt1, mt2 and dEZH2, respectively) on the structure of PRC2 (gray) and a substrate nucleosome (DNA in black, H3 histones in magenta and the other histones in pink). The structure was determined in the presence of AEBP2, JARID2 and a monoubiquitinated nucleosome (PDB 6WKR (ref. 25)), but visible are only determinants that were assayed herein, including the core PRC2 subunits and the mononucleosome. *S*-Adenosyl-homocysteine (SAH) is in CPK yellow representation. **c**, Coomassie blue-stained SDS–PAGE shows the purity of PRC2 complexes, as indicated. **d**, HMTase assay performed with various concentrations of *S*-adenosyl-methionine (SAM) (24, 12, 6, 3 and 1.5 μM on the left gel and 6, 3 and 1.5 μM on the right), using 0.6 μM wild-type or mutant PRC2 complexes. 12 μM histone H3 substrates were used in the left panel and chromatinized DNA with the sequence of a polycomb target gene

(naïve chromatin) was used in the right panel (0.645 μM chromatin used, determined based on the NCP molar equivalent). Three independent replicates were carried out on different days with similar results. **e**, HMTase assays at different PRC2 concentrations and time points, as indicated, on 1,200 nM naïve chromatin substrate. The chromatin concentration is defined as an NCP molar equivalent. Produced SAH concentrations were quantified using the MTase-Glo methyltransferase assay (Promega) and are indicated. Presented are the means of three independent replicates that were carried out on three different days and the error bar represents the standard deviation. **f**, HMTase assay performed using 0.6 μM wild-type or mutant PRC2 complexes and either 1.29 μM H3 histone or 0.645 μM chromatinized DNA substrates, with the reaction buffer supplemented with KCl to a final concentration of 0, 50 and 100 mM. Three independent replicates were carried out on different days with similar results. Uncropped gel images used to generate this figure are in Supplementary Fig. 2. Color key for all the schematic illustrations in this figure: PRC2 in gray, DNA in black, H3 histones in magenta and the other histones in pink.

that the nucleosome-interacting surface of EZH2 is required for the methylation of chromatin, even if this surface is dispensable for the methylation of unchromatinized substrates. A similar observation has been made for another EZH2 mutant, mutated in the CXC domain[24]; that mutant was active as the wild-type EZH2 on H3 peptides, exhibited only a modest reduction in affinity for nucleosomes (less than threefold dissociation constant; $K_d$) but substantially reduced methyltransferase activity on nucleosomal substrates.

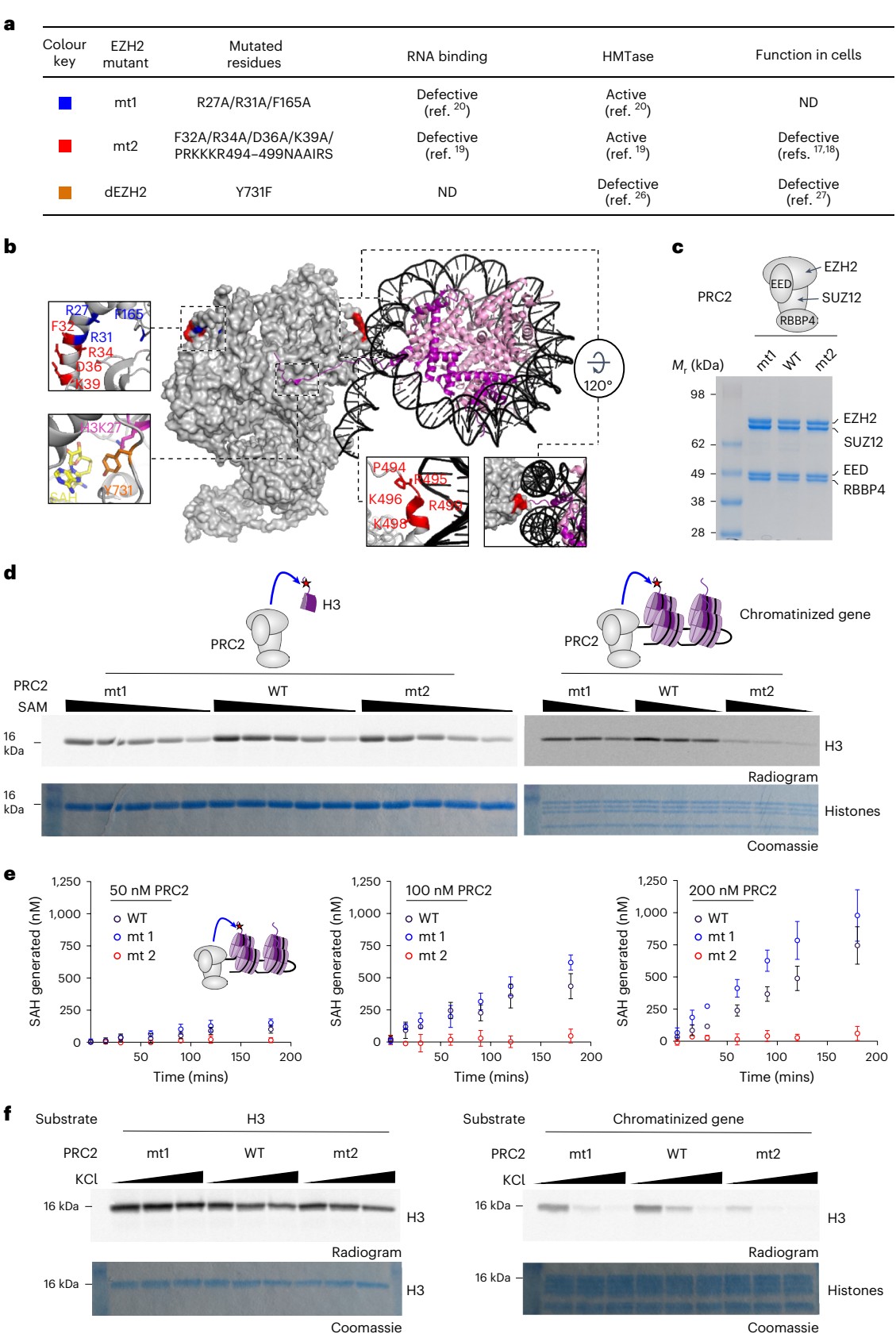

**a**

| Colour key | EZH2 mutant | Mutated residues | RNA binding | HMTase | Function in cells |
|---|---|---|---|---|---|
| 🟦 | mt1 | R27A/R31A/F165A | Defective (ref. [20]) | Active (ref. [20]) | ND |
| 🟥 | mt2 | F32A/R34A/D36A/K39A/ PRKKKR494–499NAAIRS | Defective (ref. [19]) | Active (ref. [19]) | Defective (refs. [17,18]) |
| 🟧 | dEZH2 | Y731F | ND | Defective (ref. [26]) | Defective (ref. [27]) |

When H3 histones were tested externally to the context of chromatin, the histone methyltransferase activity of the core PRC2 complex exhibited similar activity across a large range of monovalent salt concentrations (Fig. 1f, left, and Supplementary Fig. 2d). Yet, when PRC2 was tested in the context of chromatin, monovalent salt inhibits the histone methyltransferase activity of PRC2 (Fig. 1f, right, and Supplementary Fig. 2e). This indicates that electrostatic interactions are required for the methylation of chromatin by the PRC2 core complex, but not for methyltransferase per se. This strongly implies that interactions between PRC2 and nucleosomal DNA play a pivotal role during the modification of chromatin. This is in agreement with multiple high-resolution structures of PRC2 with nucleosomal constructs that identify contacts between the nucleosomal DNA and positively charged patches in surfaces of EZH2 external to the catalytic center (Fig. 1b and refs. 21,24,25).

While results thus far are internally consistent, they are not in full agreement with a previous work, which detected no difference between the methyltransferase activity of the wild-type PRC2 and mt2 when tested against a mononucleosomal construct[17]. Despite using the experimental conditions that were applied before[17], we were unable to reproduce this result. Instead, we found that the difference between the histone methyltransferase (HMTase) activities of the wild-type PRC2 and mt2 persisted regardless of which nucleosomal construct was used (Supplementary Fig. 3a,b). PRC2 mt2 was less active than the wild type regardless of whether the nucleosomes were reconstituted using the salt gradient dialysis method, as in other experiments described herein, or whether the nucleosomes were reconstituted using step dilutions, as in the original work[17] (Supplementary Fig. 3a,b). A large difference between the activity of the wild type and mt2 was also observed when PRC2 was tested on reconstituted human naïve chromatin under initial rate conditions (Fig. 1e). EZH2 mt2 also remained the least active enzyme in the presence of an allosteric-effector H3K27me3 peptide, while we noticed an elevated activity of mt1 (Supplementary Fig. 2f). Collectively, we found that mt2 exhibits lower HMTase activity, in comparison to the wild-type PRC2, against a variety of nucleosomal substrates and under different experimental conditions.

We also noticed a small reduction in the HMTase activity of mt2 with respect to the wild-type PRC2 when tested on non-nucleosomal H3 substrate at a substrate concentration of 1.2 μM (Supplementary Fig. 3a,b). These differences in activity were not as prominent as in the case of the nucleosomal substrates and were not visible at the higher concentration of histone H3 (12 μM) that was assayed herein (Fig. 1d and Supplementary Fig. 2a) and elsewhere[17].

Data thus far indicate that a portion of the RNA-binding surface of EZH2, mutated in mt2 or a portion of it (marked in red in Fig. 1b), is required for both RNA binding and for the modification of H3 histones in the context of chromatin (Fig. 1d,e and Supplementary Fig. 2b,c,e,f). Mechanistically, the interactions between the PRC2 core complex and the nucleosomal DNA are largely electrostatic (Fig. 1f) and probably involve a portion of the RNA-binding surfaces of EZH2 (Fig. 1b). A different portion of the RNA-binding surface of EZH2, represented by mt1 (marked in blue in Fig. 1b), is required for RNA binding but is dispensable for the methylation of histones, nucleosomes and chromatin (Fig. 1d and Supplementary Fig. 2a–e).

## A portion of the RNA-binding activity of EZH2 is dispensable for maintaining global H3K27me3 in cells

While both mt1 and mt2 are defective in RNA binding[19,20], they separate different sets of molecular functions: mt1 separates RNA binding from the modification of chromatin, while mt2 separates the modification of chromatin from catalysis. Given that mt2 is defective in HMTase selectively in the context of chromatin (Fig. 1d,e), we also added to our panel the catalytically defective EZH2 Y731F mutant (dEZH2; Fig. 1a,b)[26,27]. This mutation is located in the catalytic center of PRC2 (Fig. 1b, in orange)[26,27], away from all the nucleosome-binding surfaces that have been mapped

so far[21,24,25]. The mutant dEZH2 is expected to be catalytically defective against any substrate, while active in RNA binding (Supplementary Fig. 1e). We reasoned that these mutants would allow us to dissect the function of the RNA-binding activity of EZH2 from its other canonical functions (wild type (WT) versus mt1); to determine what the function of the nucleosome-interacting site is within the RNA-binding surface of EZH2 (mt1 versus mt2); and to separate the canonical HMTase activity of EZH2 in the context of chromatin from methyltransferase activity that could, in principle, take place externally to the context of chromatin and may not be limited to H3 histones (mt2 versus dEZH2).

PRC2 is required for the maintenance of transcription programs in cells that are committed to a lineage[28]. We therefore proceeded to test the different molecular functions of PRC2 in lineage-committed cells. We used the acute myelogenous leukemia (AML) K562 cell line (Fig. 2a), which is dependent on PRC2 for the maintenance of transcription programs[29]. We developed an experimental system allowing for knockout with rescue of EZH2 in bulk cell populations, to avoid variations in transcription programs that could otherwise be attributed to clonal selection. The expression level of the ectopically expressed EZH2 was controlled using flow cytometry (Fig. 2a) and was similar to that of the endogenous EZH2 (Fig. 2b and Supplementary Fig. 4a). A possible contributing factor for the even expression level of EZH2 across constructs is that the stability of the EZH2 protein is dependent on the expression level of other PRC2 core subunits[30]. Coimmunoprecipitation confirmed that all the ectopically expressed EZH2 constructs are effectively incorporated into PRC2 in the cells (Supplementary Fig. 4b). Knockout of EZH2 simultaneously with rescue using the wild-type EZH2 led to the restoration of normal H3K27me3 levels (Fig. 2b), as could be expected. Knockout of EZH2 simultaneously with rescue using dEZH2 led to global depletion of the H3K27me3 mark (Fig. 2b). There were no reproducibly substantial changes in the H3K27me1 or H3K27me2 marks (Supplementary Fig. 4a,d). We noticed an upregulation of EZH1 after EZH2 knockout, but EZH1 expression was diminished on rescue using all the EZH2 constructs (Supplementary Fig. 4a). Thus far, these results are consistent with the ectopically expressed EZH2 being the predominant EZH2 protein in PRC2 complexes and being responsible for the bulk of H3K27me3 within these cells.

EZH2 mt1 and mt2 exhibited markedly different global levels of H3K27me3: while mt1 was virtually indistinguishable from the wild type, mt2 phenocopied the catalytically defective dEZH2 (Fig. 2b and Supplementary Fig. 4a). This finding was further confirmed by repeating the assay after clonal selection of the mt1 and mt2 K562 lines (Supplementary Fig. 4c). The same observation persisted when rescue experiments were carried out in mouse embryonic stem cells (mouse ES cells), on the background of *Ezh1* and *Ezh2* knockout (Ezh1/2 dKO), using two different lentiviral expression systems (Supplementary Fig. 4e).

EZH2 mt2 includes mutations in two separate surfaces on EZH2 (Fig. 1b, marked in red). Yet, according to structural data[25], only one of the EZH2 mt2-mutated surfaces is responsible for the interactions with the substrate nucleosome (right red patch in Fig. 1b). Hence, to directly link the loss of methyltransferase activity to the substrate nucleosome-interacting surface, we generated a new mutant: mt2*. EZH2 mt2* only contains the PRKKKR494-499NAAIRS perturbation, which affects only the substrate nucleosome-interacting surface of EZH2 (right red patch in Fig. 1b). As expected, the mt2* mutant phenocopies its parental mutant mt2 in cells (Supplementary Fig. 4d). These data indicate that the global loss of H3K27me3, as seen for mt2 (Supplementary Fig. 4a), is attributed to the defective substrate nucleosome-interacting surface of EZH2 (Supplementary Fig. 4d).

Results thus far indicate that at least a portion of the RNA-binding activity of PRC2 is dispensable for the maintenance of global H3K27me3 in cells (that is, WT versus mt1 in Fig. 2b). These results also indicate that a portion of the RNA-binding surface of PRC2, which is required to modify chromatin in vitro in an RNA-independent manner, is also required to modify chromatin in cells (that is, mt1 versus mt2 in Fig. 2b).

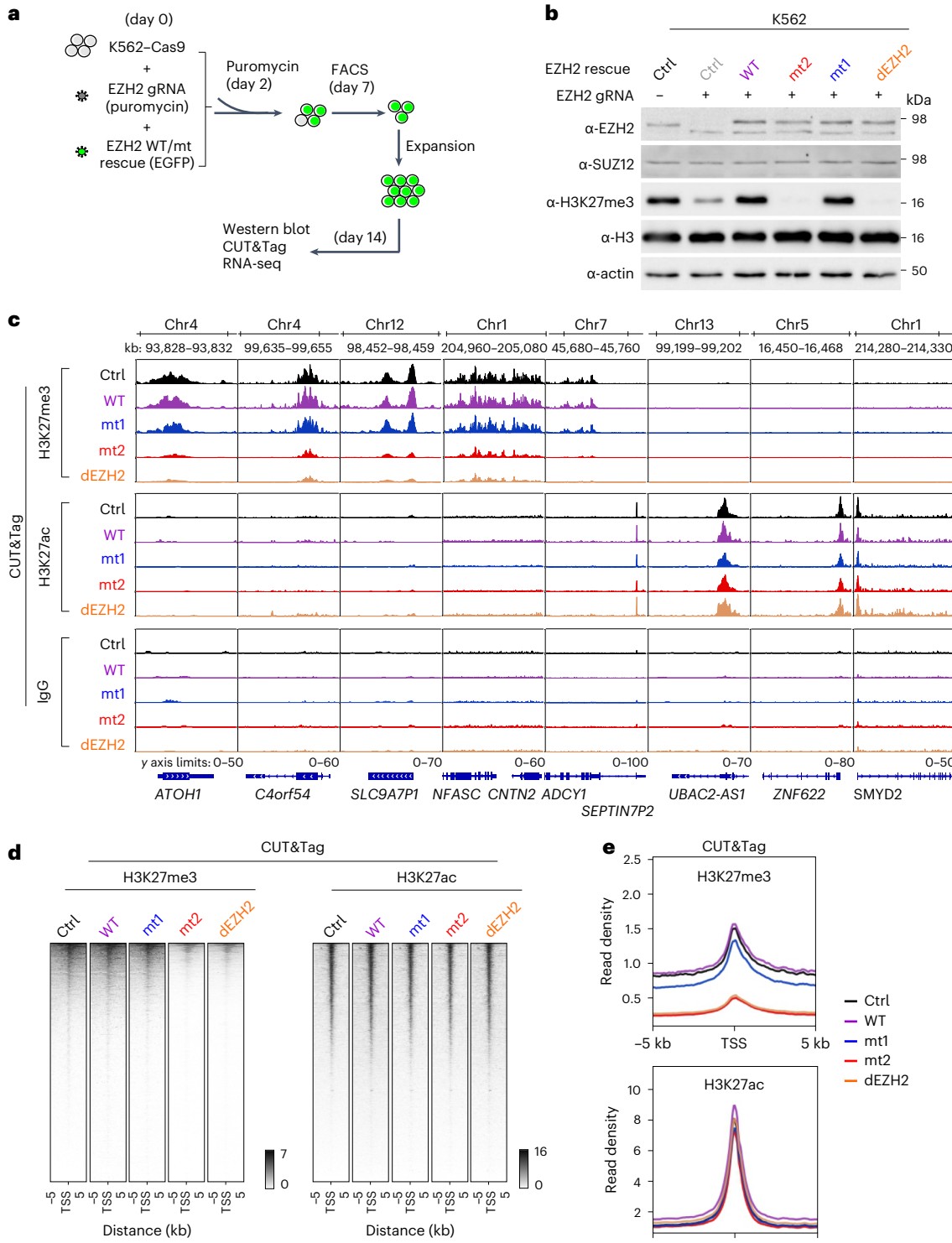

**Fig. 2 | An RNA-binding surface of EZH2, but not its RNA-binding activity, is required for maintaining H3K27me3 at repressed genes.** The control cell line (Ctrl) was transduced with the empty rescue vector and an AVVS guide RNA (gRNA) expression vector. Across the figure, the EZH2 mutants are color coded as in Fig. 1. **a**, Schematic illustration of the experimental strategy used for generating EZH2 knockout with rescue in K562 cell lines. Large circles represent cells, small circles surrounded by dots represent lentiviruses and green color represents the presence of an EGFP selection marker. **b**, Western blot analysis of whole-cell lysates collected from EZH2 knockout and rescued K562 cell lines, as indicated. The lentivirus EZH2 rescue vector is indicated, with an empty vector used as a control (Ctrl). The lentiviral gRNA vector is indicated, either if

EZH2 gRNA (+) or the AVVS gRNA control vector (−) is used. Immunoblotting antibodies are indicated on the left. Shown are from two independent replicates that were carried out on two different days, with an additional four independent replicates shown in Supplementary Fig. 2a. **c**, Representative CUT&Tag genomic tracks. **d**, Heat maps representing the H3K27me3 and H3K27ac CUT&Tag signals 5 kb upstream and downstream from all the transcription start sites (TSS) in the human genome. **e**, Enrichment profiles represent the average distributions of H3K27me3 and H3K27ac CUT&Tag over 5 kb upstream and downstream from TSS, with color code indicated. Three independent biological replicates of CUT&Tag were carried out on three different days. One replicate is presented in this figure and additional replicates are presented in Supplementary Fig. 5.

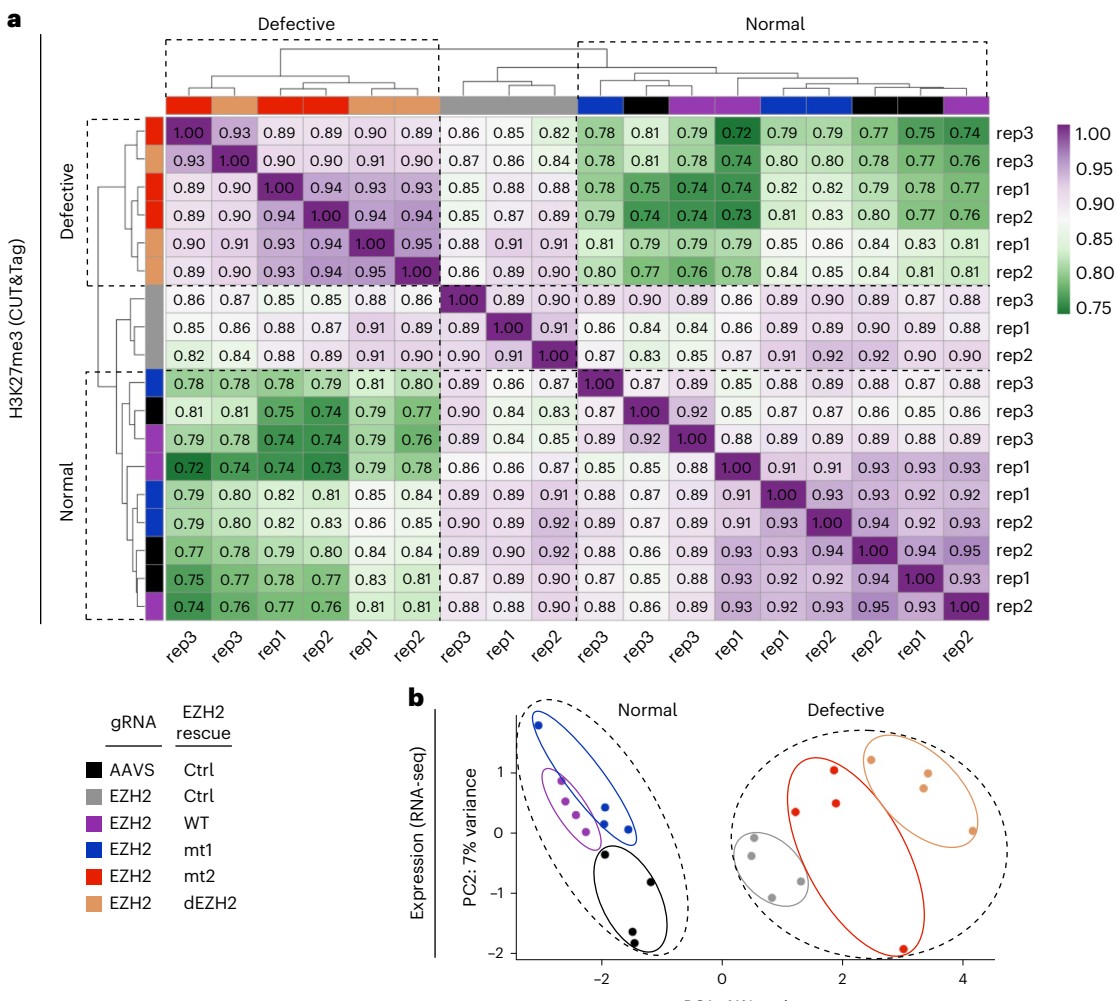

**Fig. 3 | An RNA-binding surface in EZH2, but not its RNA-binding activity, is required for maintaining transcription programs in lineage-committed cells. a**, Unsupervised clustering of genome-wide correlation between H3K27me3 CUT&Tag data from Fig. 2. Values in boxes represent Pearson's correlation coefficients between corresponding samples. Different samples are represented by color codes at the top and the left of the plot, with the color key is indicated at the bottom left of the figure. Three independent replicates were carried out on three different days, with different replicates indicated to the right and the bottom of the plot (rep). **b**, RNA-seq was carried out on cell lines that were prepared according to the same scheme as in Fig. 2a. PCA of the RNA-seq data is color coded as in panel **a**. RNA-seq recorded from four independent biological replicates that were carried out on four different days.

The data also imply that the global levels of H3K27me3 are dependent on direct contacts between nucleosomal DNA to EZH2, externally to the catalytic center and independent of catalysis (that is, mt2 versus dEZH2 in Fig. 2b).

**A portion of the RNA-binding activity of EZH2 is dispensable for depositing H3K27me3 at repressed genes**

We next aimed to determine how the different EZH2 mutants affect the deposition of H3K27me3 to repressed genes. We used the same experimental system (Fig. 2a), but instead of immunoblotting, we carried out CUT&Tag[31] using antibodies for H3K27me3 and H3K27ac (Fig. 2c–e and Supplementary Fig. 5). The level of H3K27me3 at selected target genes was similar between the control cells ('Ctrl': no knockout and no rescue) and cells where EZH2 was knocked out and rescued using the wild-type EZH2 (Fig. 2c). This trend was also similar genome wide across all transcription start sites (Fig. 2d,e) and across three biological replicates that were carried out on three different days (Fig. 2e and Supplementary Fig. 5a–c; compare black to purple enrichment profiles). By contrast, knockout and rescue with a catalytically defective EZH2 led to a substantial reduction of H3K27me3 with respect to cells expressing the wild-type EZH2 (Fig. 2d,e). Knockout of EZH2 without rescue was less effective in reducing H3K27me3 than the knockout with dEZH2 rescue (Supplementary Fig. 5c). This was possibly owing to compensation by EZH1, which exhibited elevated protein levels in the absence of EZH2 rescue (Supplementary Fig. 4a). So far, these data indicate that the knockout and rescue are effective and resemble the global levels of H3K27me3 that were observed using immunoblotting (Fig. 2b).

The two RNA-binding-defective mutants behaved differently from each other: mt1 deposited H3K27me3 at target genes as effectively as the wild-type EZH2, while mt2 resembled the catalytically defective EZH2 (Fig. 2c). Genome-wide correlation analysis of H3K27me3 levels followed by unsupervised clustering revealed two distinct clusters of cell lines, which we marked as 'defective' and 'normal' (Fig. 3a). The defective cluster included the catalytically defective EZH2 and mt2. The normal cluster included mt1, the wild-type EZH2 rescue and the control cell line that expressed the endogenous EZH2.

In contrast to the repressive H3K27me3 mark, the levels of the active H3K27ac mark remained similar between the different mutants (Fig. 2c–e and Supplementary Figs. 5a–d and 6a). This indicates that the changes in H3K27me3 deposition were directly attributed to the activity of PRC2 at its target genes, rather than to global changes in transcription or secondary effects.

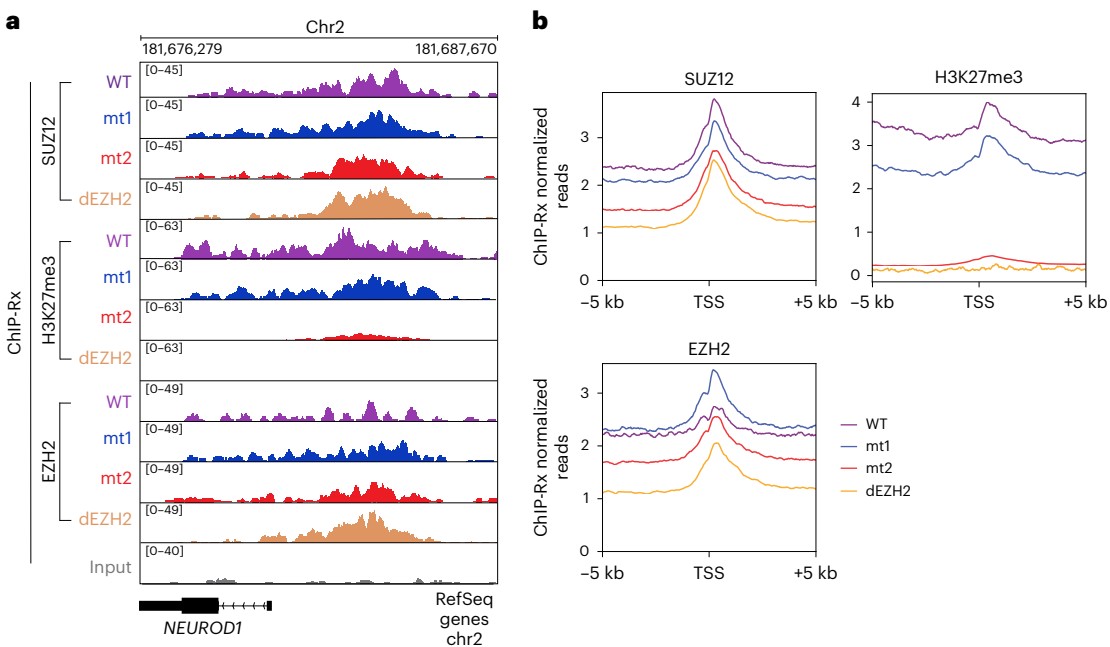

**Fig. 4 | A catalytic defective EZH2 mutation leads to lower PRC2 chromatin occupancy compared to the mutations in the nucleic acid-binding surfaces of EZH2.** **a**, Representative ChIP-Rx genomic tracks. **b**, Enrichment profiles represent the average distributions of SUZ12, EZH2 and H3K27me3 ChIP-Rx over 5 kb upstream and downstream from TSS, with color code indicated.

Data thus far indicate that the activities of mt1 and mt2 at H3K27me3-marked genes resemble their in vitro activities: mt1 is active in the modification of chromatin and mt2 is defective. This indicates that the overall RNA-binding activity of PRC2 can be reduced without compromising its ability to modify chromatin at H3K27me3-marked genes (Fig. 2; compare WT to mt1). Yet, the data also indicate that a part of the RNA-binding surface of PRC2, which also interacts with the substrate nucleosome (Fig. 1b), is required for depositing the H3K27me3 mark at target genes (Fig. 2 and Supplementary Fig. 4b, d; compare mt1 to mt2 and mt2*) and is as important as the catalytic center for H3K27me3 deposition to chromatin (Fig. 2; compare mt2 to dEZH2).

**A portion of the RNA-binding activity of EZH2 is dispensable for maintaining transcription programs in lineage-committed cells**

H3K27me3 is required for the maintenance of the repressed state of cell type-specific genes[32]. Maintaining the repressed state of cell type-specific genes enables the maintenance of transcription programs and, therefore, cell identity. Given that mt2 exhibited a rather similar loss-of-function phenotype to dEZH2 in depositing the H3K27me3 mark, we wished to determine whether these mutants also mimic each other in transcriptional regulation. For that, we used the same system (Fig. 2a) to carry out RNA-seq. K562 cells are committed to a lineage and are strongly dependent on PRC2 to maintain gene expression programs[29,33]. Accordingly, knockout with rescue using catalytically defective EZH2 led to substantial changes in transcription with respect to the wild-type rescue (Fig. 3b; compare WT to dEZH2).

Principal component analysis (PCA) on the gene expression data revealed two distinct clusters that we marked as 'normal' and 'defective' (Fig. 3b). The normal cluster included the control cells (Fig. 3b, in black), the knockout with rescue using wild-type EZH2 and the knockout with rescue using the separation-of-function mutant mt1 (Fig. 3b, in purple or blue, respectively). The defective cluster included the knockout with rescue using the catalytic defective EZH2, the knockout without rescue and the knockout with rescue using mt2 (Fig. 3b, orange, gray and red, respectively).

The dEZH2 and mt2 lines led to 1,604 and 708 differentially expressed genes, respectively, in agreement with a poor maintenance of transcription programs (Supplementary Fig. 6c). Gene ontology (GO) analysis on the differentially expressed genes identified hematopoietic-related GO terms, in agreement with the hematopoietic origin of K562 cells (Supplementary Fig. 6b). Conversely, only 57 differentially expressed genes were identified in the mt1 line, and no significant GO terms (Supplementary Fig. 6b,c). This implies that mt1 maintains transcription programs almost as the wild-type EZH2.

These results indicate that an RNA-binding-defective EZH2 can maintain transcription programs in lineage-committed cells (compare WT to mt1 in Fig. 3b). Yet, a portion of the RNA-binding surface of PRC2 is required for maintaining transcription programs (compare mt1 to mt2 in Fig. 3b), probably through interactions with nucleosomes (Fig. 1b), while PRC2 methylates them (Fig. 2). These data also suggest that a surface of EZH2 that resides externally to the catalytic center and is required for interactions with both RNA[17,19] and, independently, nucleosomal DNA[25] is as important as the catalytic center for maintaining transcription programs in lineage-committed cells (compare mt2 to dEZH2 in Fig. 3b).

**A nucleic acid-interacting surface in EZH2 is required for methyltransferase, which is in turn required for PRC2 chromatin occupancy**

We next investigated the impact of the two RNA-binding-defective EZH2 mutants on the chromatin occupancy of PRC2. To this end, we employed the same experimental system as in Fig. 2a, except that we now carried out quantitative chromatin immunoprecipitation sequencing (ChIP-seq) with exogenous reference genome spike-in (ChIP-Rx), using antibodies for H3K27me3, SUZ12 and EZH2 (Fig. 4a,b).

Consistent with the findings from the CUT&Tag analyses (Fig. 2), ChIP-Rx indicated that mt2 and dEZH2 led to a substantial reduction in H3K27me3 genome wide (Fig. 4a,b). The genome-wide chromatin occupancy of PRC2 followed a similar pattern to H3K27me3, albeit with more subtle changes: EZH2 and SUZ12 occupancy at promoter regions were slightly higher for EZH2 wild type and mt1, and slightly

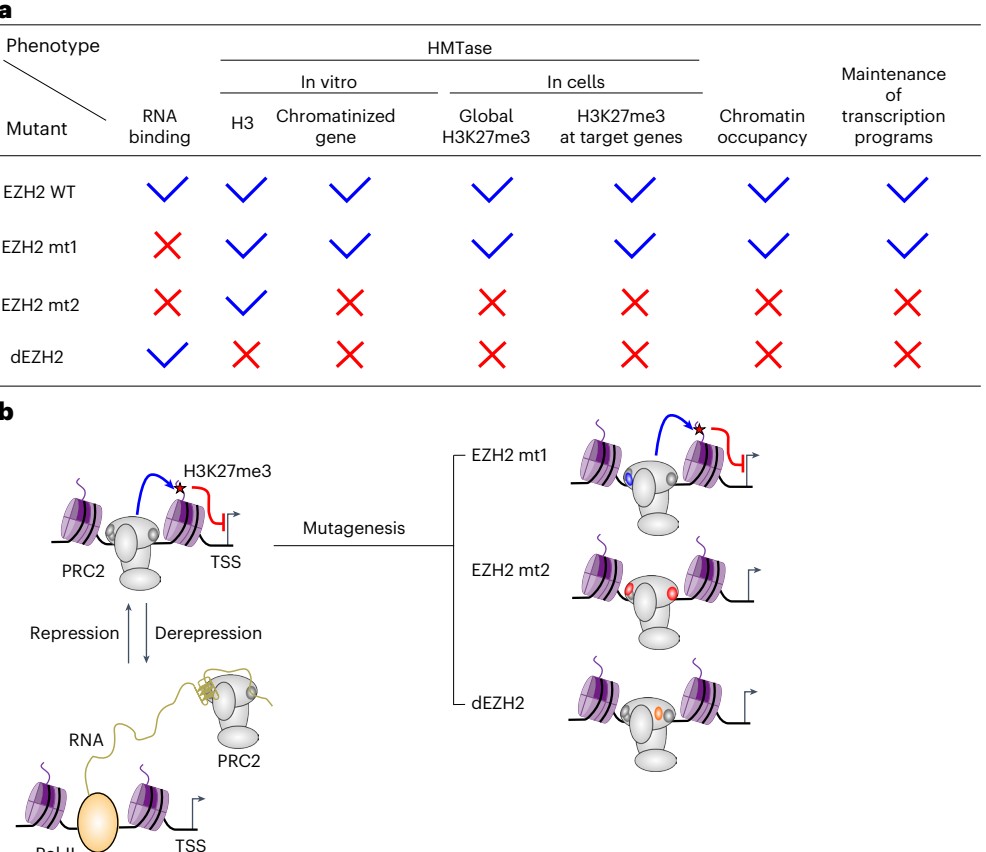

**a**

| Phenotype | HMTase | | | | | |
| | In vitro | | In cells | | | |
| Mutant | RNA binding | H3 | Chromatinized gene | Global H3K27me3 | H3K27me3 at target genes | Chromatin occupancy | Maintenance of transcription programs |
|---|---|---|---|---|---|---|---|
| EZH2 WT | ✓ | ✓ | ✓ | ✓ | ✓ | ✓ | ✓ |
| EZH2 mt1 | ✗ | ✓ | ✓ | ✓ | ✓ | ✓ | ✓ |
| EZH2 mt2 | ✗ | ✓ | ✗ | ✗ | ✗ | ✗ | ✗ |
| dEZH2 | ✓ | ✗ | ✗ | ✗ | ✗ | ✗ | ✗ |

**Fig. 5 | Model: a surface of EZH2 that can interact with either RNA or nucleosomes, rather than its RNA-binding activity per se, is required for maintaining the H3K27me3 mark at facultative heterochromatin. a**, EZH2 mutants and their identified molecular properties, as determined herein. **b**, Model: PRC2 uses part of its RNA-binding surface to interact with chromatin during histone methylation (top left). At that point, the RNA-binding surface, not its RNA-binding activity, is being used for interactions with chromatin.

When PRC2 is not methylating chromatin, EZH2 can use its RNA-binding surface to interact with RNA (bottom left). The molecular mechanism of how EZH2 mutants affect the canonical functions of PRC2 in H3K27me3 deposition (blue line) and transcriptional regulation (red line) in cells are illustrated in the top right: mt1 is indistinguishable from the wild-type EZH2, while mt2 phenocopy the catalytically defective EZH2 (dEZH2).

lower for dEZH2 and mt2 (Fig. 4a,b). These findings are consistent with the reduced PRC2 chromatin occupancy that was previously observed in the case of another catalytic defective EZH2 mutant (mouse EZH2 Y726D, equivalent to human Y731D)[34,35]. Accordingly, we also noticed a subtle increment of the soluble fraction of SUZ12 isolated from the dEZH2 and mt2 cell lines (Supplementary Fig. 7a).

Overall, these results further support that mt2 resembles the catalytic defective dEZH2. More importantly, the resemblance between the chromatin occupancies of mt2 and dEZH2 (Fig. 4) implies that the reduced chromatin occupancy identified for mt2 is more likely to be attributed to defective methyltransferase rather than to altered interactions with RNA or even nucleosomes. Hence, the substrate nucleosome-interacting surface of EZH2 (Fig. 1b) is essential for correct H3K27me3 deposition (Fig. 2), which is in turn required for transcriptional repression (Fig. 3) and the chromatin occupancy of PRC2 (Fig. 4).

## Discussion

Our data dissect different biochemical functions of EZH2 through separation-of-function mutagenesis (Fig. 5a). The conclusions are extended beyond RNA-mediated regulation. For instance, mt2 and dEZH2 (Figs. 2–4) phenocopy each other in cells. This implies that specific contacts between EZH2 and nucleosomes are as critical for the methylation of chromatin as the catalytic center itself is. Hence, although EZH2 is dispensable for the recruitment of PRC2

to chromatin[36], specific contacts between EZH2 to nucleosomes are required for histone methylation (Figs. 1–4).

The separation-of-function mutants mt1 and mt2 are not completely inactive in binding to RNA (Supplementary Fig. 1d,e and ref. 20). This is the case for all the RNA-binding-defective PRC2 mutants that were assayed by us and others[19,20]. A possible explanation is that PRC2 binds to RNA through dispersed surfaces in multiple subunits, including EZH2, EED and SUZ12 (refs. 19,20,22). Therefore, data herein cannot in any way exclude the possibility that the RNA-binding activity of PRC2 regulates, or fine tunes, some of its functions. Yet, mt1 reduces the affinity of PRC2 for RNA by over twofold $K_d$, according to data presented here (Supplementary Fig. 1d,e) and elsewhere[20]. Despite this, mt1 is biochemically (Fig. 1) and phenotypically (Figs. 2–4) nearly indistinguishable from the wild-type EZH2. This indicates that the full extent of the RNA-binding activity of EZH2 is not required for the canonical functions of PRC2: H3K27 methylation of chromatin and the maintenance of transcription programs.

How does the RNA-binding surface of EZH2 operate? The RNA-binding-defective mt2 modifies H3 histones only when they are isolated, but not in the context of nucleosomes or naïve chromatin (Fig. 1). These observations have nothing to do with the RNA-binding activity of PRC2, as all the in vitro methyltransferase assays herein (Fig. 1 and Supplementary Figs. 1–3) were carried out in the absence of RNA. In fact, our RNA-binding assays indicate that EZH2 mt2 is quite able to interact with RNA (Supplementary Fig. 1d,e), despite being

defective in chromatin modification (Fig. 1e). Hence, a portion of the RNA-binding surface of PRC2 is required for the methylation of H3 histones in the context of chromatin and independently of RNA binding. This observation provides a mechanistic explanation to rationalize previous findings, indicating that PRC2 can either bind to nucleosomes or to RNA, but not to both of them simultaneously[13,14]. When PRC2 is bound to chromatin, it uses part of its RNA-binding surface to engage with nucleosomes and these interactions are required for the methylation of chromatin and are independent of RNA (Fig. 5b, left). The interactions between EZH2 and nucleosomes are also required for the chromatin occupancy of PRC2 (Fig. 4), but this could be attributed to the H3K27me3 deposition (Fig. 4; compare mt2 to dEZH2). When PRC2 is not bound to chromatin, part of its chromatin-interacting surface is required for RNA binding (Fig. 5b, right).

A portion of the RNA-binding surface of EZH2 is required for the function of PRC2 in human pluripotent stem cells[17,18]. The same RNA-binding surface is also required for the canonical functions of PRC2 in cells, including H3K27me3 deposition in lineage-committed cells and stem cells and the maintenance of gene expression programs (Figs. 2–4 and Supplementary Figs. 4–6), but also for the modification of chromatin in vitro (Fig. 1). Given the data herein, the simplest explanation for seemingly contradictory observations in the recent literature, implicating RNA as either a negative[10–14] or a positive[17,18] regulator of PRC2, is as follows: a nucleosome-interacting surface of EZH2, not the RNA-binding activity of this surface, is required for the canonical functions of PRC2 through engaging with chromatin during histone methylation.

## Online content

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

## Methods

### Protein expression and purification

The full-length human EZH2, SUZ12, RBBP4 and EED (UniProtKB: Q15910-2, Q15022-1, Q09028-1 and O75530-1, respectively) were cloned into the pFastBac1 expression vector containing a PreScission-cleavable N-terminal hexahistidine-MBP tag, as previously described[7,37]. Mutations were introduced to EZH2 using Takara PrimeSTAR HS (Clontech, cat. no. R045A) or Pfu DNA polymerase, as previously described[20]. Baculovirus production, titration, infection and cell collection, and the purification of PRC2 and the mutants, were carried out as previously described[20]. All the proteins were snap-frozen in liquid nitrogen and stored at −80 °C as single-use aliquots.

To resolve all PRC2 subunits using SDS–PAGE, 3–8% Tris–acetate gel (Thermo Fisher, cat. no. EA0375BOX) and 1× MES SDS running buffer (Thermo Fisher, cat. no. NP0002) were used. The 3 µg PRC2 complexes were supplemented to a final concentration of 1× LDS sample buffer (Thermo Fisher, cat. no. NP0007) with 1% 2-mercaptoethanol (Sigma-Aldrich, cat. no. M3148) and heated at 95 °C for 5 min before loading onto a Tris–acetate gel. The gel was run for 30 min at 200 V before staining with InstantBlue Coomassie protein stain (Expedeon, cat. no. ISB1L).

### Nucleosome reconstitution

Unlabeled and Cy5-labeled nucleosomes were produced as previously described[29]. In brief, recombinant human histones were purified from inclusion bodies and reconstituted into histone octamers as previously described[38]. Cy5-labeled H2A was produced as previously described[39]. The 147-Base pair (bp) or 182-bp DNA including one copy of the 601 Widom sequence was PCR amplified and purified using anion exchange column (Cytiva, cat. no. 17115401) with NaCl gradient from 150 mM to 2 M. Mononucleosomes were reconstituted by initially titrating across three octamer ratios (from 1:0.8 to 1:1.2 DNA:octamer molar ratio) in a 20-µl mixture of 6 µM DNA, 20 mM Tris pH 7.5, 2 M KCl, 1 mM EDTA, 10 mM DTT. For each of these samples, gradient salt dialysis was used at 4 °C in a dialysis device (Thermo Fisher, cat. no. 69572), starting from refolding buffer (20 mM Tris pH 7.5, 2 M KCl, 1 mM EDTA, 1 mM DTT) to a medium salt buffer containing 20 mM Tris pH 7.5, 250 mM KCl, 1 mM EDTA and 1 mM DTT over 18 h, and the final-step dialysis was carried out using a low salt buffer containing 20 mM Tris pH 7.5, 2.5 mM KCl, 1 mM EDTA, 1 mM DTT. Quality of mononucleosomes was assessed by 5–6% acrylamide TBE gel electrophoresis, and the most appropriate molar ratio of DNA:octamer was selected for large-scale reconstitution. Large-scale reconstitution was conducted as above, except in a volume of 0.2–2 ml using dialysis tubing (Spectrum, cat. no. 888-11527). Mononucleosomes were stored at 4 °C and the quality was assessed by 5–6% acrylamide TBE gel.

For mononucleosomes reconstituted using step dilutions, we followed the procedure as previously described[17]. In brief, a 300-µl starting reaction was prepared by mixing 1.95 nmol of DNA with a 1.2 molar ratio of histone octamers (2.34 nmol) in starting buffer (2 M NaCl, 6 mM Tris pH 7.5, 0.3 mM EDTA and 0.3 mM DTT). The mixture was first incubated for 30 min at 37 °C. Dilution buffer (20 mM Tris pH 7.5, 1 mM EDTA, 1 mM DTT) was then added to the mixture in 30-min intervals in the following volumes: 324, 360, 840 and 1,920 µl. Mononucleosomes were then concentrated with Amicon Ultra-0.5 ml 10-kDa cutoff centrifugal filter (Merck, cat. no. UFC501096) and stored at 4 °C.

Chromatinized genes were produced as previously described[29]. Briefly, ATOH1 DNA was amplified using Pfu DNA polymerase and purified by ion exchange chromatography. Purified DNA was concentrated by isopropanol precipitation and was then dissolved in Tris–EDTA buffer. Chromatin was assembled using the gradient salt dialysis at 4 °C. Chromatin was reconstituted by initially titrating across a range of octamer ratios (from 1:12 to 1:24 DNA:octamer molar ratio) in a 20-µl mixture of 0.1–0.3 µM DNA, 20 mM Tris pH 7.5, 2 M KCl, 1 mM EDTA and 10 mM DTT. For each of these samples, gradient salt dialysis was used at 4 °C in a dialysis device (Thermo Fisher, cat. no. 69572), starting from refolding buffer (20 mM Tris pH 7.5, 2 M KCl, 1 mM EDTA and 1 mM DTT) to a medium salt buffer containing 20 mM Tris pH 7.5, 250 mM KCl, 1 mM EDTA and 1 mM DTT over 18 h, and the final-step dialysis was carried out using a low salt buffer containing 20 mM Tris pH 7.5, 2.5 mM KCl, 1 mM EDTA, 1 mM DTT. Quality of chromatinized genes was assessed by 0.8% agarose TBE gel electrophoresis, and the most appropriate molar ratio of DNA:octamer was selected for large-scale reconstitution. Large-scale reconstitution was conducted as above, except in a volume of 0.2–2 ml using dialysis tubing (Spectrum, cat. no. 888-11527). To concentrate the assembled chromatin, MgCl₂ was added to a final concentration of 20 mM. Then, the mixture was incubated for 15 min at room temperature, followed by 15 min on ice. The mixture was then centrifuged at 4 °C for 20 min at 20,000 relative centrifugal force, and precipitate was resuspended in the low salt buffer. The concentration of the nucleosome core particles in the arrays was measured using BCA assay (Thermo Fisher, cat. no. 23252). Chromatinized genes were stored at 4 °C and the quality was assessed by 0.8 % agarose TBE gel electrophoresis.

### Micrococcal nuclease digestion

Micrococcal nuclease (MNase) digestion was performed as previously described[40] with some changes. Specifically, DNA was diluted to 75 ng µl⁻¹ and chromatinized ATOH1 was diluted to 150 ng µl⁻¹ (DNA concentration) in a buffer containing 25 mM HEPES–KOH pH 7.5, 10% glycerol, 100 mM KCl, 3 mM MgCl₂, 1 mM EDTA and 1 mM DTT. Three MNase (NEB, cat. no. M0247S) dilutions were made in a buffer containing 10 mM HEPES pH 7.5, 10 mM KCl, 1.5 mM MgCl₂ and 10% glycerol to final concentrations of 45 U µl⁻¹, 15 U µl⁻¹ and 5 U µl⁻¹. Then, 5 µl of the respective MNase dilution was added to 35 µl of the samples and the reaction was started by adding 5 µl of CaCl₂ from a 10 mM stock to give a final concentration of 2 mM. The reaction proceeded at room temperature for 7 min. The reaction was stopped by adding 5 µl EDTA from a 500 mM stock, followed by 100 µl of glycogen stop buffer (20 mM EDTA, 200 mM NaCl, 1% SDS, 0.25 mg ml⁻¹ glycogen). Samples were digested by adding 1 µl of Proteinase K (NEB, cat. no. P8107S) from a 20 mg ml⁻¹ stock followed by incubation for 30 min at 37 °C. The DNA fragments were purified using a MinElute PCR cleanup kit (Qiagen, cat. no. 28004). After purification, samples were incubated at 55 °C with opened lids for 5 min to evaporate residual ethanol. DNA fragments were separated on a 1.2% agarose gel in TAE buffer and visualized with GelRed.

### Analytical ultracentrifugation

All sedimentation velocity experiments were performed using an Optima analytical ultracentrifuge (Beckman Coulter) at a temperature of 20 °C. Sedimentation velocity runs were recorded by measuring absorbance at 280 or 275 nm at a rotor speed of 12,000 rpm.

Chromatinized ATOH1 and ATOH1 DNA were diluted in 25 mM Tris pH 7.5, 10 mM KCl, 1 mM EDTA and 1 mM TCEP to final concentrations of 0.025 µM and 41.6 ng µl⁻¹, respectively. Experiments were performed in a conventional double-sector quartz cell. Solvent density (0.9996 g ml⁻¹ at 20 °C) and viscosity (1.0100 cP at 20 °C) as well as octamer partial specific volume (0.7443 ml g⁻¹) were computed using the program SEDNTERP[41]. The ATOH1 DNA partial specific volume (0.5 ml g⁻¹) was adopted from the reported values for DNA[42]. The specific volumes of octamers and ATOH gene were used to compute the partial specific volume for ATOH arrays (0.65 ml g⁻¹), as previously described[43,44]. Finally, the sedimentation velocity data were fitted to a continuous size (c(s)) using the program SEDFIT[45].

### In vitro HMTase activity assays using radiolabeled SAM

HMTase activity assays were performed as previously described[17], with some modifications. In brief, for the HMTase reactions with histone H3 as substrates, each 10-µl reaction contained 0.6 µM PRC2, 12 µM

H3.1 and five concentrations of S-[methyl-$^{14}$C]-adenosyl-L-methionine (PerkinElmer, cat. no. NEC363050UC; twofold serial dilutions, starting from 24 µM) were incubated in the reaction buffer A (50 mM Tris–HCl pH 8.0 at 30 °C, 100 mM KCl, 2.5 mM MgCl$_2$, 0.1 mM ZnCl$_2$, 2 mM 2-mercaptoethanol and 0.1 mg ml$^{-1}$ BSA, 5% v/v glycerol) for 1 h at 30 °C.

For the 10-µl reactions with mononucleosomes or nucleosomal arrays as substrates, 0.6 µM PRC2, 0.6 µM mononucleosomes or 70 ng µl$^{-1}$ nucleosomal arrays, and three concentrations (6, 3 and 1.5 µM) of $^{14}$C-labeled SAM were incubated in reaction buffer B (50 mM Tris–HCl pH 8.5 at 30 °C, 5 mM MgCl$_2$ and 4 mM DTT) for 2 h at 30 °C. For the assays performed at 0 mM, 50 mM and 100 mM KCl, 10-µl reactions contained 0.6 µM PRC2, 6 µM S-[methyl-$^{14}$C]adenosyl-L-methionine, either 0.645 µM nucleosomal arrays (where the molar concentration is defined as NCP equivalent) or 1.29 µM H3.1 and reaction buffer B with either 0 mM KCl, 50 mM KCl or 100 mM KCl final concentration.

For the HMTase reactions comparing mononucleosomes that were produced using different reconstitution methods (that is, gradient dialysis and step dilution), each 10-µl reaction contained 0.6 µM PRC2, 1.2 µM H3 or 0.6 µM mononucleosomes (NCP$_{147}$ or NCP$_{182}$ that were reconstituted either by step dilutions or gradient dialysis) and three concentrations (24, 12 and 6 µM) of S-[methyl-$^{14}$C] adenosyl-L-methionine. All reactions were assayed in a reaction buffer of 50 mM Tris pH 8.5 at 30 °C, 5 mM MgCl$_2$ and 4 mM DTT, and incubated at 30 °C for 2 h. All the reactions were stopped by adding 4× LDS loading dye (Thermo Fisher Scientific, cat. no. NP0007) to a final concentration of 1× LDS with 1% 2-mercaptoethanol (Sigma-Aldrich, cat. no. M3148) and heating at 95 °C for 5 min. The reactions were then loaded onto 16.5% SDS–PAGE gels and run on ice for 120 min at 160 V. Gels were stained with InstantBlue Coomassie protein stain (Expedeon, cat. no. ISB1L) before vacuum drying for 1 h at 80 °C. Dried gels were then exposed to a storage phosphor screen for several days before acquiring radiograms using a Typhoon 5 Imager (GE Healthcare). All experiments were performed in three independent replicates that were carried out on three different days. Densitometry was carried out using ImageQuant software (GE Healthcare). Relative activity of nucleosomal substrates was obtained by dividing all densitometry values of the same replicate by the densitometry value of H3 methylation of NCP$_{182}$ (gradient dialysis) by PRC2 wild-type protein at the highest SAM concentration of a given replicate. Relative activity in the presence of the H3 substrate was obtained by dividing all densitometry values within a given replicate by the densitometry value for the same substrate as quantified in the presence of the PRC2 wild-type protein at the highest SAM concentration. This was performed for all three replicates for all substrates. The resulting values were then plotted with error bars using GraphPad Prism 9 software.

### Time-course in vitro methyltransferase-Glo assay

Before the MTase-Glo assay, 5 µM PRC2 was incubated with 10 µM PALI1-K1241me3 peptide and 20 µM SAM for 30 min at 30 °C in the HMTase buffer (50 mM Tris pH 8.0 at 30 °C, 0.5 mM MgCl$_2$, 0.1% Tween-20, 5 mM DTT and 35 mM KCl). Then, various concentrations of 50, 100 or 200 nM PRC2 and 1,200 nM *ATOH1* nucleosomal arrays (concentration defined as NCP molar equivalent) were assayed with 25 µM SAM in the HMTase buffer in a 384-well plate (Sigma, cat. no. M3561). In experiments with an allosteric-effector peptide, H3K27me3 peptide was added to a final concentration of 50 µM. For each enzyme, either wild type or mutant, a separate reaction without nucleosomal substrate was set for background subtraction. Reactions were incubated at 30 °C and quenched using 1 µl of 2.45% v/v TFA at 10 time points between 0 to 180 min. Standard curves were created for each replicate using twofold serial dilutions of SAH from 1 µM to 15.6 nM in the same buffer as the samples. The luminescence signal was developed using the MTase-Glo methyltransferase assay kit (Promega, cat. no. V7602) and captured using a BMG FLUOstar OPTIMA plate reader (BMG Labtech). Three independent measurements were performed on three different days.

### RNA-binding assays using fluorescence anisotropy

Fluorescence anisotropy (FA) was performed as previously described[20]. Briefly, 3′ fluorescein labeled G4 24 RNA (UUAGGG)4 was incubated for 2 min at 95 °C in 10 mM Tris–HCl pH 7.5 and was then snap-cooled on ice for 2 min. RNA was then incubated for 30 min at 37 °C in binding buffer (50 mM Tris–HCl pH 7.5 at 25 °C, 200 mM KCl, 2.5 mM MgCl$_2$, 0.1 mM ZnCl$_2$, 2 mM 2-mercaptoethanol, 0.1 mg ml$^{-1}$ BSA (NEB, cat. no. B9000S), 0.05% Nonidet P40 (Roche, cat. no. 11754599001) and 0.1 mg ml$^{-1}$ fragmented yeast tRNA (Sigma, cat. no. R5636)). After that, RNA was combined with serial dilutions of the protein for a final reaction volume of 20 µl containing 5 nM fluorescently labeled RNA at the desired final protein concentration. The mixture was equilibrated at 30 °C for 30 min before measurement. Fluorescence anisotropy data were collected using a PHERAstar plate reader (BMG Labtech) at 30 °C ($\lambda_{ex}$ = 485 nm, $\lambda_{em}$ = 520 nm). The background was subtracted from protein-free samples. $K_d$, Hill and standard error values were calculated with GraphPad Prism 9 software using nonlinear regression for specific binding with Hill slope function.

For assaying the RNA-binding activities using different RNAs, the following RNA probes were synthesized as 3′ 6-FAM-labeled RNA by IDT: HOTAIR[46], MEG3 (ref. [47]) the G4 RNA probe described above and a size-matched G4 mt RNA without G tracts[20]. FA was performed as described above, except that the binding buffer contained 100 mM KCl. The probe sequences were as follows:

HOTAIR RNA: GGGAGCCCAGAGUUACAGACGGCGGCGAGAG GAAGGAGGGGCGU
MEG3 RNA: UGCCCAUCUACACCUCACGAGGGCA
G4 RNA: (UUAGGG)4
G4 mt RNA: (UGAGUG)4

### Electrophoretic mobility shift assay

EMSA was carried out as previously described[29]. Briefly, PRC2 dilutions and Cy5-H2A labeled mononucleosomes (final concentration 5 nM) were incubated at 4 °C for 30 min in binding buffer (50 mM Tris–HCl, pH 7.5 at 25 °C, 100 mM KCl, 2 mM 2-mercaptoethanol, 0.05% v/v NP-40, 0.1 mg ml$^{-1}$ BSA, 5% glycerol). The reaction mixtures were then subjected to nondenaturing gel electrophoresis at 6.6 V cm$^{-1}$ over a 0.7% agarose gel buffered with 1× TBE at 4 °C for 30 min. The Cy5 dye signals were captured using a Typhoon 5 Imager. Three independent replicates were carried out on three different days.

### Cell culture

K562 cells were cultured in RPMI 1640 (Merck, cat. no. R8758) growth medium supplemented with 10% FBS (Cellsera, cat. no. AU-FBS/SF) and 1% (v/v) penicillin–streptomycin (Thermo Scientific, cat. no. 15140122) and were incubated at 37 °C with 5% CO$_2$. Mouse ES cells were cultured on 0.1% gelatin-coated dishes in DMEM medium supplemented with 20% FBS, 1% (v/v) penicillin–streptomycin, 50 µM β-mercaptoethanol, 1:100 GlutaMAX (Thermo, cat. no. 35050061), 1:100 MEM nonessential amino acids (Thermo, cat. no. 11140076), 1:100 sodium pyruvate (Thermo, cat. no. 11360070), 1:1,000 leukemia inhibitory factor (LIF; produced in-house), 3 µM CHIR99021 (STEMCELL, cat. no. 72054) and 1 µM PD0325901 (STEMCELL, cat. no. 72184). HEK293T cells (for lentiviral production) were cultured in DMEM medium supplemented with 10% FBS, 1% (v/v) penicillin–streptomycin. K562 cells were acquired from ATCC and all cells were tested periodically for mycoplasma contamination.

### CRISPR–Cas9 knockout of EZH2 and rescue in K562 cells

*EZH2* gRNA sequence (AATAATCAGGCATACCATCT) and a negative control gRNA targeting *AAVS1* locus (CGGGCCCCTATGTCCACTTC) were subcloned into BsmBI linearized pXPR_003 vector (puromycin selection). Flag-EZH2 WT and mutants were subcloned into SmaI (NEB, cat. no. R0141) linearized pHIV-EGFP (Addgene, cat. no. 21373, EGFP selection using a polycistronic expression system) vector using Gibson

Assembly and NEB Stable Competent *E. coli* (NEB, cat. no. C3040). The plasmids were fully sequenced. Lentiviruses were generated and stored as previously described[29]. For the generation of Cas9-expressing K562 cells, plasmids for the polycistronic expression of Cas9 and mCherry were packed into lentiviruses and were transduced into K562 cells. The cells were selected using flow cytometry. For EZH2 depletion and rescue experiments, $3 \times 10^4$ K562 cells were transduced in RPMI and 8 µg ml$^{-1}$ polybrene using 200 µl lentivirus stock of either *EZH2* gRNA or *AAVS1* gRNA, and 200 µl lentivirus stock of either rescue or empty vector to a final volume of 700 µl in a treated 24-well plate (day 0). On day 2, half of the cell suspension was moved to a 10-cm dish with puromycin added to a final concentration of 10 µg ml$^{-1}$. On day 7, cells were sorted by flow cytometry on the basis of EGFP and mCherry signals and seeded into a 10-cm dish. On day 14, cells were trypsinized, washed twice with PBS by centrifugation at 500 relative centrifugal force for 5 min and were then resuspended in PBS to 1 million per ml before aliquoting them for the downstream applications, as described below. This entire process, starting from day 0, was carried out independently for each replicate while starting on a different day. For experiments done using the derived cell lines, see Supplementary Methods.

### Generation of Ezh1/2 dKO and EZH2 rescue mouse ES cell lines

For subcloning of human EZH2 WT or mutants into a pLenti expression vector, full-length open reading frames (ORFs) of EZH2 WT or mutants were PCR amplified and subcloned into the pCR™8/GW/TOPO Gateway cloning entry vector according to the instructions of the manufacturer (Thermo, cat. no. K250020). The EZH2 ORFs were subsequently subcloned into Gateway destination vector pLENTI-EFIA-FLAG/HA (a gift from A. Bracken, Trinity College Dublin) using Clonase Gateway LR Clonase II Plus enzyme (Thermo, cat. no. 12538120) according to the instructions of the manufacturer. The plasmids were fully sequenced. Lentiviruses were generated using HEK293T cells as described above.

The *Ezh1/2* dKO cell line was generated by treating the *Ezh1*$^{-/-}$; *Ezh2*$^{f/f}$;*Rosa26CreERT2* mouse ES cell line (generated by the laboratory of K. Helin[36] with 0.5 µM 4-OHT (Sigma, cat. no. H7904) for 96 h. For generating the human EZH2 pLenti rescues in the *Ezh1/2* dKO cell line, 150,000 mouse ES cell were seeded per well on a six-well plate, one day before the viral transduction. The mouse ES cells were then treated with virus for 48 h before puromycin selection at 1 µg ml$^{-1}$. For generating the human EZH2 pHIV-EGFP rescues in the *Ezh1/2* dKO cell line, the same lentiviral transduction procedure was applied and cells were sorted using flow cytometry on the basis of the EGFP signal after 48 h treatment with virus. For experiments done using the derived cell lines, see Supplementary Methods.

### Reporting summary

Further information on research design is available in the Nature Portfolio Reporting Summary linked to this article.

### Data availability

CUT&Tag, RNA-seq and ChIP-Rx data and processed files have been deposited in the NCBI GEO database under accession number GSE239447. Source data are provided with this paper.

### Code availability

Data analyses were performed with standard published software, as described in Methods, using scripts that are available via Zenodo at https://doi.org/10.5281/zenodo.10866993 (ref. 48).

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

### Acknowledgements

We thank the support of the Monash FlowCore and the MASSIVE HPC facility. Q.Z. is supported by Investigator Grant EL1 (APP1196365) from National Health and Medical Research Council (NHMRC). C.D. is an EMBL-Australia Group Leader and a Sylvia and Charles Viertel Senior Medical Research Fellow and acknowledges support from the ARC (DP190103407) and the NHMRC (APP1162921, APP1184637, APP2011767 and APP2020900). This research was funded partially by the Victoria State Government through mRNA Victoria to Q.Z. and C.D.

### Author contributions

Q.Z., S.F.F., E.H., N.J., M.U., V.L. and X.H.N. carried out experiments. Q.Z., E.H.G., S.F.F., E.H. and X.H.N. analyzed data. Q.Z., E.H.G., E.H. and C.D. wrote the paper. Q.Z., E.H.G., E.H. and C.D. designed the project. Q.Z. and C.D. supervised the project.

### Competing interests

The authors declare no competing interests.

### Additional information

**Correspondence and requests for materials** should be addressed to Qi Zhang or Chen Davidovich.

# Reporting Summary

## Statistics

For all statistical analyses, confirm that the following items are present in the figure legend, table legend, main text, or Methods section.

| n/a | Confirmed | |
|---|---|---|
| ☐ | ☒ | The exact sample size (*n*) for each experimental group/condition, given as a discrete number and unit of measurement |
| ☐ | ☒ | A statement on whether measurements were taken from distinct samples or whether the same sample was measured repeatedly |
| ☐ | ☒ | The statistical test(s) used AND whether they are one- or two-sided *Only common tests should be described solely by name; describe more complex techniques in the Methods section.* |
| ☒ | ☐ | A description of all covariates tested |
| ☒ | ☐ | A description of any assumptions or corrections, such as tests of normality and adjustment for multiple comparisons |
| ☐ | ☒ | A full description of the statistical parameters including central tendency (e.g. means) or other basic estimates (e.g. regression coefficient) AND variation (e.g. standard deviation) or associated estimates of uncertainty (e.g. confidence intervals) |
| ☐ | ☒ | For null hypothesis testing, the test statistic (e.g. *F*, *t*, *r*) with confidence intervals, effect sizes, degrees of freedom and *P* value noted *Give P values as exact values whenever suitable.* |
| ☒ | ☐ | For Bayesian analysis, information on the choice of priors and Markov chain Monte Carlo settings |
| ☒ | ☐ | For hierarchical and complex designs, identification of the appropriate level for tests and full reporting of outcomes |
| ☐ | ☒ | Estimates of effect sizes (e.g. Cohen's *d*, Pearson's *r*), indicating how they were calculated |

*Our web collection on statistics for biologists contains articles on many of the points above.*

## Software and code

Policy information about availability of computer code

| Data collection | Amersham Typhoon Scanner Control software 1.1, PHERAstar 4.00R4, ChemiDoc™ imager, BD Influx™ cell sorter, HiSeq X10, Novaseq 6000 |
|---|---|
| Data analysis | GraphPad Prism 9, SEDNTERP; SEDFIT (16.1c); GraphPad Prism 9; Bowtie2 (2.2.9 and 2.3.5); Picard toolkit (2.26.2); ChIPseqSpikeInFree R package (1.2.4); deepTools (3.1.3 and 3.3.0); pyGenomeTracks (3.5); pheatmap (version 1.0.12); SEACR (1.3); ChIPpeakAnno R package (3.22.4); Salmon (0.14.1); DESeq2 (1.28.1); tximport (1.16.1); limma (3.44.3); clusterProfiler (3.16.1); SAMtools (1.3.1 and 1.9-gcc5); ImageQuant (TL v8.1.0.0.) |

For manuscripts utilizing custom algorithms or software that are central to the research but not yet described in published literature, software must be made available to editors and reviewers. We strongly encourage code deposition in a community repository (e.g. GitHub). See the Nature Portfolio guidelines for submitting code & software for further information.

## Data

Policy information about availability of data

All manuscripts must include a data availability statement. This statement should provide the following information, where applicable:

- Accession codes, unique identifiers, or web links for publicly available datasets
- A description of any restrictions on data availability
- For clinical datasets or third party data, please ensure that the statement adheres to our policy

CUT&Tag, RNA-Seq and ChIP-Rx data and processed files have been deposited in the NCBI GEO database under accession number GSE239447.

# Research involving human participants, their data, or biological material

Policy information about studies with <u>human participants or human data</u>. See also policy information about <u>sex, gender (identity/presentation), and sexual orientation</u> and <u>race, ethnicity and racism</u>.

| | |
|---|---|
| Reporting on sex and gender | N/A |
| Reporting on race, ethnicity, or other socially relevant groupings | N/A |
| Population characteristics | N/A |
| Recruitment | N/A |
| Ethics oversight | N/A |

Note that full information on the approval of the study protocol must also be provided in the manuscript.

# Field-specific reporting

Please select the one below that is the best fit for your research. If you are not sure, read the appropriate sections before making your selection.

☒ Life sciences ☐ Behavioural & social sciences ☐ Ecological, evolutionary & environmental sciences

For a reference copy of the document with all sections, see <u>nature.com/documents/nr-reporting-summary-flat.pdf</u>

# Life sciences study design

All studies must disclose on these points even when the disclosure is negative.

| | |
|---|---|
| Sample size | All experiments were carried out in at least two or three replicates to ensure independent experiments are reproducible, with the exception of ChIP-Rx that was carried out in one replicate. ChIP-Rx were sequenced to a depth of 20 million reads, which is sufficient for analyzing the chromatin binding of PRC2 and the localization of H3K27me3. |
| Data exclusions | No data were excluded from the analyses. |
| Replication | See figure legends for information about the number of independent replicates that were carried out for each of the experiments. |
| Randomization | Randomization was not carried out because different samples included different reagents (e.g. proteins, substrates, antibodies, buffers, etc.) or were subjected to different treatments as indicated, which complicated randomization. Yet, experiments were designed to include controls that allows for objective interpretation of the results. |
| Blinding | Not performed, as the experiments and data analysis are not subjective. |

# Reporting for specific materials, systems and methods

We require information from authors about some types of materials, experimental systems and methods used in many studies. Here, indicate whether each material, system or method listed is relevant to your study. If you are not sure if a list item applies to your research, read the appropriate section before selecting a response.

## Materials & experimental systems

| n/a | Involved in the study |
|---|---|
| ☐ | ☒ Antibodies |
| ☐ | ☒ Eukaryotic cell lines |
| ☒ | ☐ Palaeontology and archaeology |
| ☒ | ☐ Animals and other organisms |
| ☒ | ☐ Clinical data |
| ☒ | ☐ Dual use research of concern |
| ☒ | ☐ Plants |

## Methods

| n/a | Involved in the study |
|---|---|
| ☐ | ☒ ChIP-seq |
| ☒ | ☐ Flow cytometry |
| ☒ | ☐ MRI-based neuroimaging |

# Antibodies

| | |
|---|---|
| Antibodies used | anti-Actin (Sigma #A2066, 1:500), anti-EZH2 (Active Motif #39875 for WB with 1:5000 dilution; Cell Signaling #5246 for ChIP-Rx, 6.3 |

| Antibodies used | µg per ChIP), anti-H3 (Abcam #ab1791, 1:50000), anti-H3K27me3 (Cell signaling #9733 for WB with 1:4000 dilution and for CUT&Tag with 1:50 dilution; Cell Signaling #35861SF for ChIP-Rx, 5 µg per ChIP; Active Motif #61017 for WB with 1:2500 dilution), anti-SUZ12 (Santa Cruz Biotechnology #sc-271325 for WB with a dilution of 1:200, Cell signaling #3737S for co-IP with 1.5 µg per IP and ChIP-Rx with 0.5 µg per ChIP), anti-mouse HRP-conjugated (Jackson Immuno-Research #715-035-150, 1:5000), anti-rabbit HRP-conjugated (Santa Cruz Biotechnology #sc2357, 1:5000), anti-H3K27ac (Abcam #4729, 1:50), IgG control (Cell signaling #2729S for CUT&Tag with 1:50 dilution, Cell signaling #3900S for co-IP with 1.5 µg per IP), Guinea pig anti-rabbit antibody (Antibodies online #ABIN101961, 1:100), anti-CBX7 (Abcam #21873, 1:1000), anti-EZH1 (Cell Signaling #42088, 1:1000), anti-H3K27me1 (Merck #07-448, 1:1000), anti-H3K27me2 (Abcam #24684, 1:2000), anti-GAPDH (Proteintech #10494-1-AP, 1:4000). |
|---|---|
| Validation | All antibodies were commercially available and validated by the manufacture. Anti-Actin (Sigma #A2066) antibody has been validated by Sigma using enhanced antibody validation assay. Anti-EZH2 (Active Motif #39875) antibody has been validated by knocking out EZH2 in our lab, as shown in Fig. 2b (compare the first two lanes of EZH2 blot). The Cell signaling EZH2 antibody (CST #5246) has been validated by Cell Signaling using SimpleChIP Enzymatic Chromatin IP Kit. Anti-H3 antibody (Abcam #ab1791) has been validated by Abcam using western blot. Anti-H3K27me3 (Cell signaling #9733 and #35861SF) were produced from the same clone C36B11, and have been validated by Cell Signaling using SimpleChIP Enzymatic Chromatin IP Kit. Anti-H3K27me3 (Active Motif #61017) has been validated by Active Motif using dot blot. anti-SUZ12 (Santa Cruz Biotechnology #sc-271325) has been validated by Santa Cruz using western blot. anti-SUZ12 (Cell signaling #3737S) has been validated by Cell Signaling using SimpleChIP Enzymatic Chromatin IP Kit. Anti-mouse HRP-conjugated (Jackson Immuno-Research #715-035-150) has been validated by Jackson Immuno-Research using ELISA and/or solid-phase adsorbed. Anti-H3K27ac (Abcam #ab4729) has been validated by Abcam using western blot and ChIP. Guinea pig anti-rabbit antibody (Antibodies online #ABIN101961) has been tested by Antibodies online using ELISA and immunoelectrophoresis assay. Both IgG control (CST #2729 and #3900) have been validated by Cell Signaling using SimpleChIP Enzymatic Chromatin IP Kit. anti-CBX7 (Abcam #ab21873) has been validated by Abcam using western blot. anti-EZH1 (CST #42088) has been validated by Cell Signaling using western blot. anti-H3K27me1 (Merck #07-448) has been tested by Merck using western blot. anti-H3K27me2 (Abcam #24684) has been tested by Abcam using western blot. anti-GAPDH (Proteintech #10494-1-AP) has been validated by Proteintech using western blot, ELISA, FC, IF, IHC, IP. |

# Eukaryotic cell lines

Policy information about cell lines and Sex and Gender in Research

| Cell line source(s) | K562 cells were purchased from ATCC by our lab. HEK293T cells were a gift from Jose Polo lab, Monash University (unknown commercial source). Ezh1-/-;Ezh2f/f;Rosa26CreERT2 mESC line was generated by the lab of Kristian Helin. |
|---|---|
| Authentication | K562 cells were obtained from ATCC and authenticated by ATCC. HEK293T and Ezh1-/-;Ezh2f/f;Rosa26CreERT2 mESC line were not authenticated. |
| Mycoplasma contamination | All cell lines are routinely tested for mycoplasma contamination using PCR. The cells used in this work tested negative. |
| Commonly misidentified lines (See ICLAC register) | No commonly misidentified cell lines were used. |

# Plants

| Seed stocks | *Report on the source of all seed stocks or other plant material used. If applicable, state the seed stock centre and catalogue number. If plant specimens were collected from the field, describe the collection location, date and sampling procedures.* |
|---|---|
| Novel plant genotypes | *Describe the methods by which all novel plant genotypes were produced. This includes those generated by transgenic approaches, gene editing, chemical/radiation-based mutagenesis and hybridization. For transgenic lines, describe the transformation method, the number of independent lines analyzed and the generation upon which experiments were performed. For gene-edited lines, describe the editor used, the endogenous sequence targeted for editing, the targeting guide RNA sequence (if applicable) and how the editor was applied.* |
| Authentication | *Describe any authentication procedures for each seed stock used or novel genotype generated. Describe any experiments used to assess the effect of a mutation and, where applicable, how potential secondary effects (e.g. second site T-DNA insertions, mosiacism, off-target gene editing) were examined.* |

# ChIP-seq

## Data deposition

☒ Confirm that both raw and final processed data have been deposited in a public database such as GEO.

☒ Confirm that you have deposited or provided access to graph files (e.g. BED files) for the called peaks.

| Data access links *May remain private before publication.* | https://www.ncbi.nlm.nih.gov/geo/query/acc.cgi?acc=GSE239447 |
|---|---|
| Files in database submission | GSM5974936 Ctrl-Ctrl_IgG_cuttag_rp1<br>GSM5974937 Ctrl-Ctrl_H3K27me3_cuttag_rp1<br>GSM5974938 Ctrl-Ctrl_H3K27ac_cuttag_rp1<br>GSM5974939 KO-Ctrl_IgG_cuttag_rp1<br>GSM5974940 KO-Ctrl_H3K27me3_cuttag_rp1<br>GSM5974941 KO-Ctrl_H3K27ac_cuttag_rp1 |

GSM5974942 KO-WT_IgG_cuttag_rp1
GSM5974943 KO-WT_H3K27me3_cuttag_rp1
GSM5974944 KO-WT_H3K27ac_cuttag_rp1
GSM5974945 KO-mt2_IgG_cuttag_rp1
GSM5974946 KO-mt2_H3K27me3_cuttag_rp1
GSM5974947 KO-mt2_H3K27ac_cuttag_rp1
GSM5974948 KO-mt1_IgG_cuttag_rp1
GSM5974949 KO-mt1_H3K27me3_cuttag_rp1
GSM5974950 KO-mt1_H3K27ac_cuttag_rp1
GSM5974951 KO-dEZH2_IgG_cuttag_rp1
GSM5974952 KO-dEZH2_H3K27me3_cuttag_rp1
GSM5974953 KO-dEZH2_H3K27ac_cuttag_rp1
GSM5974954 Ctrl-Ctrl_IgG_cuttag_rp2
GSM5974955 Ctrl-Ctrl_H3K27me3_cuttag_rp2
GSM5974956 Ctrl-Ctrl_H3K27ac_cuttag_rp2
GSM5974957 KO-Ctrl_IgG_cuttag_rp2
GSM5974958 KO-Ctrl_H3K27me3_cuttag_rp2
GSM5974959 KO-Ctrl_H3K27ac_cuttag_rp2
GSM5974960 KO-WT_IgG_cuttag_rp2
GSM5974961 KO-WT_H3K27me3_cuttag_rp2
GSM5974962 KO-WT_H3K27ac_cuttag_rp2
GSM5974963 KO-mt2_IgG_cuttag_rp2
GSM5974964 KO-mt2_H3K27me3_cuttag_rp2
GSM5974965 KO-mt2_H3K27ac_cuttag_rp2
GSM5974966 KO-mt1_IgG_cuttag_rp2
GSM5974967 KO-mt1_H3K27me3_cuttag_rp2
GSM5974968 KO-mt1_H3K27ac_cuttag_rp2
GSM5974969 KO-dEZH2_IgG_cuttag_rp2
GSM5974970 KO-dEZH2_H3K27me3_cuttag_rp2
GSM5974971 KO-dEZH2_H3K27ac_cuttag_rp2
GSM5974972 Ctrl-Ctrl_IgG_cuttag_rp3
GSM5974973 Ctrl-Ctrl_H3K27me3_cuttag_rp3
GSM5974974 Ctrl-Ctrl_H3K27ac_cuttag_rp3
GSM5974975 KO-Ctrl_IgG_cuttag_rp3
GSM5974976 KO-Ctrl_H3K27me3_cuttag_rp3
GSM5974977 KO-Ctrl_H3K27ac_cuttag_rp3
GSM5974978 KO-WT_IgG_cuttag_rp3
GSM5974979 KO-WT_H3K27me3_cuttag_rp3
GSM5974980 KO-mt2_IgG_cuttag_rp3
GSM5974981 KO-mt2_H3K27me3_cuttag_rp3
GSM5974982 KO-mt2_H3K27ac_cuttag_rp3
GSM5974983 KO-mt1_IgG_cuttag_rp3
GSM5974984 KO-mt1_H3K27me3_cuttag_rp3
GSM5974985 KO-mt1_H3K27ac_cuttag_rp3
GSM5974986 KO-dEZH2_IgG_cuttag_rp3
GSM5974987 KO-dEZH2_H3K27me3_cuttag_rp3
GSM5974988 KO-dEZH2_H3K27ac_cuttag_rp3
GSM5974989 Ctrl-Ctrl_RNAseq_rp1
GSM5974990 KO-Ctrl_RNAseq_rp1
GSM5974991 KO-WT_RNAseq_rp1
GSM5974992 KO-mt2_RNAseq_rp1
GSM5974993 KO-mt1_RNAseq_rp1
GSM5974994 KO-dEZH2_RNAseq_rp1
GSM5974995 Ctrl-Ctrl_RNAseq_rp2
GSM5974996 KO-Ctrl_RNAseq_rp2
GSM5974997 KO-WT_RNAseq_rp2
GSM5974998 KO-mt2_RNAseq_rp2
GSM5974999 KO-mt1_RNAseq_rp2
GSM5975000 KO-dEZH2_RNAseq_rp2
GSM5975001 Ctrl-Ctrl_RNAseq_rp3
GSM5975002 KO-Ctrl_RNAseq_rp3
GSM5975003 KO-WT_RNAseq_rp3
GSM5975004 KO-mt2_RNAseq_rp3
GSM5975005 KO-mt1_RNAseq_rp3
GSM5975006 KO-dEZH2_RNAseq_rp3
GSM5975007 Ctrl-Ctrl_RNAseq_rp4
GSM5975008 KO-Ctrl_RNAseq_rp4
GSM5975009 KO-WT_RNAseq_rp4
GSM5975010 KO-mt2_RNAseq_rp4
GSM5975011 KO-mt1_RNAseq_rp4
GSM5975012 KO-dEZH2_RNAseq_rp4
GSM7665216 1-K562-IN
GSM7665217 2-WT-SUZ12
GSM7665218 3-mt1-SUZ12
GSM7665219 4-mt2-SUZ12
GSM7665220 5-dEZH2-SUZ12

GSM7665221 6-WT-EZH2
GSM7665222 7-mt1-EZH2
GSM7665223 8-mt2-EZH2
GSM7665224 9-dEZH2-EZH2
GSM7665225 10-WT-K27me3
GSM7665226 11-mt1-K27me3
GSM7665227 12-mt2-K27me3
GSM7665228 13-dEZH2-K27me3

Genome browser session
(e.g. UCSC)

No longer applicable

## Methodology

Replicates

3 independent replicates for CUT&Tag  and 1 replicate for ChIP-Rx

Sequencing depth

2.5-23x10^6 reads per sample for CUT&Tag, 2-3x10^7 reads per sample for ChIP-Rx

Antibodies

CUT&Tag: H3K27me3: Cell Signaling Technology #9733; H3K27ac: Abcam #4729; Rabbit IgG control: Cell Signaling Technology
#2729; Guinea pig anti-rabbit antibody: Antibodies online #ABIN101961. Primary antibodies: 1:50, secondary antibody: 1:100
ChIP-Rx: Anti-EZH2 (Cell Signalling #5246), 6.3 µg per ChIP. Anti-SUZ12 (Cell Signalling #3737), 0.5 µg per ChIP. Anti-H3K27me3 (Cell
Signalling #35861SF), 5 µg per ChIP.

Peak calling parameters

No peaks were called from ChIP-seq data.

Data quality

No peaks were called from ChIP-seq data.

Software

Bowtie2 (2.2.9 and 2.3.5); Picard toolkit (2.26.2); ChIPseqSpikeInFree R package (1.2.4); deepTools (3.1.3 and 3.3.0);
pyGenomeTracks (3.5); pheatmap (version 1.0.12); SEACR (1.3); ChIPpeakAnno R package (3.22.4); Salmon (0.14.1); DESeq2 (1.28.1);
tximport (1.16.1); limma (3.44.3); clusterProfiler (3.16.1); SAMtools (1.3.1 and 1.9-gcc5)

