## [Peer Review File · Nature Genetics]

Peer Review Information

Manuscript Title: Inseparable RNA binding and chromatin modification activities of a nucleosome-interacting surface in EZH2

Corresponding author name(s): Dr Chen Davidovich, Dr Qi Zhang

Reviewer Comments & Decisions:

Decision Letter, initial version:

20th Aug 2022

Dear Chen,

Your Article entitled "An RNA binding surface in EZH2 is required for the deposition of H3K27me3 to chromatin in an RNA-independent manner" has now been seen by 4 referees, whose comments are attached. I apologize for the long review process.

After careful consideration and in light of the reviewers' advice we have decided that we cannot offer to publish your manuscript in Nature Genetics.

While the referees find your work of some interest, they raise concerns about the appropriateness of some of the technical approaches, and the strength of the novel conclusions that can be drawn at this stage.

Reviewer #1 thinks this manuscript could potentially bring some clarity to the ongoing PRC2-RNA debate but it would require a major revision. Amongst other things, you would need to test the binding of the EZH2 mutants to different types of RNA, the cell type used (K562) is not ideal (please see below), and you'd need to gain more additional mechanistic insight to fill in some gaps. Reviewer #2 says the data are of high quality in general, but they think that many key conclusions are not well supported. The reviewer has several experimental suggestions for potential improvement. Reviewer #3 feels that the findings are potentially interesting, but that additional controls are required. Like Reviewer #1, they think that the RNA-binding ability of these mutants would need to be tested more in-depth and since Long et al. used pluripotent stem cells, for a fair comparison, you should do the same. Reviewer #4 thinks that some of the key biochemical assays would need to be redone and that the EZH2's RNA-binding properties would need to be more comprehensively characterized. They therefore are skeptical about the current conclusions.

We feel that these reservations are sufficiently important as to preclude publication of this study in Nature Genetics.

I am sorry that we cannot be more positive on this occasion but hope that you will find our referees' comments helpful when preparing your paper for submission elsewhere.

Sincerely,

Tiago

Tiago Faial, PhD
Chief Editor
Nature Genetics
<https://orcid.org/0000-0003-0864-1200>

Reviewers' Comments:

Reviewer #1:

Remarks to the Author:

PRC2 ability to bind RNA has been recently suggested to be essential for PRC2 activity, maintenance of gene expression and proper iPS cells differentiation (Long et al., 2020). In this work, Zhang and colleagues assess in vitro and in vivo activity of two reported RNA binding-defective EZH2 mutants. They provide convincing evidence for the fact that at least one of the two mutations affects PRC2 activity due to impaired chromatin interaction in an RNA-independent manner, thus providing a framework for the reinterpretation of previous results (Long et al., 2020). In its simplicity, the work represents an essential piece of the PRC2-RNA puzzle that definitely deserves to be shared with the community. Yet, several data presented as main figures, could be moved to supplementary figures, and the overall manuscript can be formatted in a 2/3 figures piece.

Nevertheless, this manuscript has two main limitations, which should be addressed experimentally:

1) Binding to RNAs

The ability of EZH2 (wild-type and mutated forms) to bind RNAs is only tested with respect to a single family of RNAs (i.e. G4 24 RNA: UUAGGGx4). The authors should extend this analysis to other types of RNAs. It cannot be excluded that point mutations might generate an affinity for other RNA classes. This is particularly relevant since other PRC2 subunits might retain (or even gain) affinity for RNA. Thus PRC2-RNA binding should be tested in vivo, preferably using orthogonal approaches.

2) K562 bulk cell populations

Analysis of K562 bulk population limits significantly the interpretation of the results. The authors should repeat some of the key experiments using selected clones of K562 or moving to hiPS or mESCs.

Main points:

1) The authors never analyzed (or mentioned) the effects of mt1 and mt2 mutants on H3K27me1 and H3K27me2.

2) Are the amino-acids mutated in mt1 and mt2, conserved in other species. This might be important

to understand when the affinity for nucleosomal DNA/RNAs appear during evolution?

3) Which are the levels of EZH1 in K562 cells? Is there any compensation of EZH1 in cells expressing mt1 and mt2 mutants?

4) The authors should perform cell fractionation to investigate whether wt, mt1, mt2, dEZH2 have different localization/affinity for chromatin.

5) In acute myelogenous leukemia cells, overexpression of PALI1 leads to cell differentiation. The authors should include similar experiments to assess the impact of gene expression changes (upon expression of EZH2 mutants) on the ability of cell to differentiate.

6) As mentioned above, RNA-seq analysis is based on the assumption that PRC2-mediated gene regulation depends on the presence of EZH2. However, Figure 2b and S2d show that the endogenous EZH2 protein is variably expressed. To rule out the possibility that endogenous EZH2 might be supporting mt1 mutant in aiding PRC2-mediated gene silencing, the authors should repeat the analysis of the transcriptome using selected clones for each genotype.

7) Provided the data generated for transcriptome analysis, the paper would benefit from a more thorough analysis of the differentially expressed genes other than the sole PCA analysis presented in Figure 4b (e.g. providing numbers and overlap with ChIP-seq peaks of H3K27me3, correlation between loss of H3K27me3 and transcription upregulation, overlap with dEZH2), GO analysis of differentially expressed genes etc... .

Minor points:

- I would suggest to remove replicates from main figure (e.g. Figure 2b) or merge into single plots where possible (eg. fig. 3b,d)
- Provide a bar-plot quantification to summarize WB data of Figure 2
- Figure 2 and 3 could be easily merged (without replicates)
- Fig. 4 is not so relevant; it should be moved to supplementary figures.
- Fig. 5a should be reformatted

Typos:

Fig. 1a 'Function' in cells

L226 across all transcription start 'sited'

Reviewer #2:

Remarks to the Author:

A) In the manuscript entitled "An RNA binding surface in EZH2 is required for the deposition of H3K27me3 to chromatin in an RNA-independent manner" Zhang and colleagues try to connect the RNA binding surface of EZH2 to its catalytic activity both in vitro and in vivo. In addition, their findings support the ability of EZH2 (and the PRC2 in general) to bind and modify chromatin in a RNA-independent manner.

The main point of this study consists in understanding whether the RNA binding ability of EZH2/PRC2 is involved in targeting the PRC2 to chromatin and that affects its catalytic ability to deposit H3K27 methylation.

B) The major contribution and advancement produced by this study is the "reclassification" of an EZH2 mutant previously considered catalytic active while RNA-binding deficient (termed mt2 in this manuscript and previously characterized in PMID: 32632336) to a catalytic impaired mutant specifically toward nucleosomal incorporated H3. As the authors pointed out, their results disagree with previous findings reported using very similar conditions concerning the activity of the mt2 mutant

(PMID: 32632336).

This study identifies an important feature of EZH2, and PRC2 in general, that is required for chromatin binding and catalytic activity of the complex. I am less convinced the experiments and results presented ultimately solved the issue of whether PRC2 requires RNA binding to be targeted to chromatin and exert its catalytic function.

C) Quality of the data and reproducibility by the authors are excellent. Data presentation is good however including concentrations, gradients of all the reagents in each figure would highly improve and speed up interpretations of the figures by the readers.

D) Statistical assessment of the data is appropriate; some figures might require appropriate quantification and/or statistical assessment (Fig. S1B-C)

E) Data appear robust, reliable and valid across figures and panels

F) Major points:

1) Based on the data presented by the authors, the best summary of their findings is perfectly represented by the sentence at line 122-125 "This also indicates that the nucleosome-interacting surface of EZH2 is required for the methyltransferase of chromatin, even if this surface is dispensable for the methyltransferase of unchromatinized substrates." The title and the abstract that refer mainly to the EZH2 RNA binding ability, however the most important data in the manuscript point out to EZH2 and its ability to interact/modify nucleosomes. Both the mutants presented do not massively lose their ability to bind RNA (Fig. S1B, especially mt2 Vs wt). However, the effect of the mt2 EZH2 compared to WT (or to mt1 EZH2) on PRC2 chromatin binding/methyltransferase activity *ex vivo* (in the cells) and *in vitro* (nucleosomes) is remarkable. Therefore, it could be concluded that PRC2 containing mt2 EZH2 is defective in chromatin binding despite being quite able to interact with RNA. This could, potentially, be even further demonstrated by measuring the RNA binding ability of the dEZH2 the authors used in this study (there should be no reason why this mutant should not interact with RNA). This supports that EZH2 RNA binding ability is not sufficient for PRC2 targeting to chromatin/methyltransferase activity but does not answer whether it is necessary.

2) Is there any RNA copurified with the PRC2 or the nucleosomes in the *in vitro* assays? It would be good to include a step with RNase A and/or RNase H in the key *in vitro* experiments (nucleosome interaction and methyltransferase assays). That would help the authors to understand if there is any RNA contribution to the effect they see *in vitro*. The main question is can the PRC2 associate to chromatin and modify it without any RNA present in the assay?

3) An important mutant the authors should consider introducing in the present study is the EZH2 PRKKKR494-499NAAIRS without mutations is F32/R34/D36/K39. Based on the cryo-EM structure and the authors' data the interface nucleosomal DNA/EZH2 is very important for PRC2 binding to chromatin and for its catalytic activity. However, based on the cryo-EM structure, only the 6aa PRKKKR in EZH2 are part of the interface. Therefore, including a third mutant (mt3?) could strengthen the authors conclusions, narrow down the effects described by the authors to a more precise EZH2 protein region and improve the interpretation of the data.

4) The most puzzling results that complicate the interpretation of the study come from the NCP interaction experiments in Fig. S1C. When these results are taken into consideration to interpret all the other results, it is not clear the mechanism behind the effects described by the authors. A simple explanation for the entire study, as mentioned by the authors (Line 122-125), is that they identified a nucleosomal DNA binding region (mediated or not by RNA, answering my question #2 will help to figure this out) of EZH2 that is important for PRC2 binding to chromatin and for its catalytic function. However, both mt1 EZH2 and mt2 EZH2 have the same defective interaction with NCPs (Fig. S1C, I

would also invite the authors to quantify these interactions). At this point it would be logical to conclude that the interaction with NCPs is quite dispensable for PRC2 binding to chromatin and for its catalytic activity. This seems unrealistic. However, the conditions used in the EMSA and the methyltransferase assays are different (accordingly to the M&M section).

Can the two assays be performed using comparable concentrations of nucleosome and PRC2?

Is it possible that in the EMSA assays the two different mutants can separately bind nucleic acids (DNA and/or RNA) but while the mt1 EZH2 loses not specific binding ability and therefore not strictly required for PRC2 functions, the mt2 loses important binding ability required for PRC2 interaction and modification of NCPs? The authors should investigate this in more details since it seems to be crucial to their data interpretation/explanation. So far, they just try to explain this important issue in lines 104-106 "This led us to hypothesize that mt2 might not be able to interact with chromatin in a conformation required for the efficient modification of histone tails".

It would be interesting competition assays with non-specific DNA/RNA and examine how the two mutants behave in EMSA and HMT assays.

It would be important to use the chromatinized gene (ATO1) in the EMSA studies.

5) The rescue experiments in Figure 2 are convincing, however, I would recommend the authors to introduce the mutations by CRISPR especially if the mt3 (see point 3) can recapitulate the effects of mt2. This would be a cleaner system that would avoid co-existence of WT/truncated EZH2 with the EZH2 versions expressed by the rescue constructs.

G) References are appropriate

H) The manuscript is generally well written, however the constant reference to RNA binding ability and RNA binding surface is quite confusing. Also, the authors might want to reconsider the title that is also not very explanatory. The authors might want to reconsider changing the "RNA binding surface" to "nucleosome binding interface/surface/module" especially if they can further experimentally clarify the points at (F).

Minor comments:

Figure 1A has a typo "functiuon in cells"

Reviewer #3:

Remarks to the Author:

In this manuscript, Chen and colleagues addressed the role of RNA-binding in mediating the establishment of H3K27 methylation by PRC2. As the authors stated in the introduction, this is a key and controversial issue, with some recent studies suggesting that RNA-binding is essential for PRC2 recruitment. Here the authors performed biochemical analysis and found that one of the RNA-binding defective mutant EZH2 (mt2) in fact loses the ability to interact with nucleosomal DNA and is unable to methylate H3K27 when mono-nucleosomes and nucleosome arrays are used as substrates. In contrary, another RNA-binding defective mutant EZH2 (mt1), retains the methyltransferase activity. By expressing these mutants in EZH2 knockout K562 cells, authors conclude that EZH2 mt1 but not mt2 is essentially undistinguishable with wildtype EZH2 in terms of restoring levels of H3K27me3 and gene expression, suggesting that RNA-binding activity per se is not required for PRC2 function.

Overall this is an interesting study. I have the following suggestions mainly to enhance the study rigor. While some of the control experiments may seem excessive, given the provocative nature of the conclusion I believe they should be included.

1. The RNA-binding activity of EZH2 mt1 and mt2 needs further assessment. Only one assay was performed and shown in Fig. S1 with somewhat modest reduction in binding. Authors need to include additional assays (eg. EMSA) and test greater diversity of synthetic RNAs (eg. G-quadruplex-forming RNA) to ensure that mt1 is in fact defective in RNA-binding. Even better is to show that EZH2 mt1 mutant binding to RNA is reduced in cells.
2. Quality control data is missing for the in vitro reconstituted chromatin using ATOH1 locus. What is the density and distribution patterns of nucleosomes on this synthetic DNA template?
3. Given that PRC2 activity can be allosterically activated by H3K27me3, could authors test the effect of spiking in H3K27me3 peptides or nucleosomes on the methyltransferase activity of mt1 and mt2 EZH2?
4. In the publication by Long et al. 2020 which is heavily referred to in this manuscript, human pluripotent stem cells were used for cellular functional studies. Here authors used K562 cells. For a fair comparison, could the authors study mt1 and mt2 EZH2 in a stem cell context?
5. In CUT&Tag studies, only parental K562 was included as control. K562 cells expressing EZH2 gRNA should also be included as additional control.
6. In addition to H3K27me3, authors should perform (1) co-IP to test the PRC2 complex assembly with EZH2 mt1 and mt2 in cells and (2) ChIP to examine genome-wide binding patterns of EZH2 mt1 and mt2. These studies would be quite informative. Related, in Long et al. 2020 study, ChIP for PRC2 was performed with and without treatment of RNAse, I wonder if RNAase treatment would affect the recruitment of EZH2 mt1 mutant?
7. In previous publications by the author's group, it was proposed that promiscuous RNA binding by PRC2 represents a mechanism to exclude PRC2 recruitment to transcriptionally active regions. According to this model, EZH2 mt1 would be predicted to localize to at least some active genes? Can authors analyze the H3K27me3 data to see if this is true?

Reviewer #4:
Remarks to the Author:
Main concerns:

1. How many RNA-binding domains are there within EZH2? How do the authors know that they studied the relevant RNA-binding domain?
2. There are different assays used utilizing different substrates, octamers (or H3), which is irrelevant as PRC2 is not active with core histones or octamers, mononucleosomes and oligonucleosomes. The conditions for each assay are different, but most importantly, when analyzing enzymes and comparing

wild-type to mutants, the assays must be performed during the initial part of the reaction (rate), not incubated for 2 hrs (yield in real biochemical terms).

3. The most surprising aspect to me is that the authors appear to forget a basic principle of biochemistry, when comparing different mutants, as is in this study, one must use different concentrations of proteins and compare their activity, and the assay must be performed under rate conditions. None of these basic concepts of biochemistry were applied in this study. Thus, I do not believe the conclusions.

Decision Letter, Appeal:

7th Oct 2022

Dear Chen,

Thank you for asking us to reconsider our decision on your manuscript entitled "An RNA binding surface in EZH2 is required for the deposition of H3K27me3 to chromatin in an RNA-independent manner".

I have now discussed the points of your letter with my colleagues, and we think that you have some valid points. We therefore invite you to revise your manuscript along the lines that you propose.

However, we feel it's important to point out that both reviewers #1 and #3 would ideally like to see some key experiments redone with human pluripotent stem cells. Without doing this, one cannot definitively rule out that some of the apparent differences between your findings and those of Long et al. are due to distinct biological contexts (PSCs vs. K562 cells). In the absence of these experiments, the conclusions will need to be very carefully worded.

As per the reviewers' points, we expect to see a careful and comprehensive testing of the different mutants' abilities to bind distinct types of RNA.

Finally, we'd like to stress that even if Long et al. performed some experiments that are potentially flawed or limited in their design, this does not justify not performing new experiments to a higher standard, particularly when highlighted by a reviewer (e.g. reviewer #4's points). We will ask reviewer #4 to comment on the revised manuscript again regarding the suitability of the biochemical assays (Supp. Fig. 1).

When preparing a revision, please ensure that it generally complies with our editorial requirements for format and style; details can be found in the Guide to Authors on our website (<http://www.nature.com/ng/>).

Please be sure that your manuscript is accompanied by a separate letter detailing the changes you have made and your response to the points raised. At this stage we will need you to upload:

- 1) a copy of the manuscript in MS Word .docx format.
- 2) The Editorial Policy Checklist:

<https://www.nature.com/documents/nr-editorial-policy-checklist.pdf>

3) The Reporting Summary:

(Here you can read about the role of the Reporting Summary in reproducible science:

<https://www.nature.com/news/announcement-towards-greater-reproducibility-for-life-sciences-research-in-nature-1.22062>)

Please use the link below to be taken directly to the site and view and revise your manuscript:

[redacted]

With kind wishes,

Tiago

Tiago Faial, PhD

Chief Editor

Nature Genetics

<https://orcid.org/0000-0003-0864-1200>

Author Rebuttal to Initial comments

We would like to thank the reviewers for dedicating their time to review our manuscript, we believe their suggestions allowed us to improve the manuscript substantially.

20th Aug 2022

Dear Chen,

Your Article entitled "An RNA binding surface in EZH2 is required for the deposition of H3K27me3 to chromatin in an RNA-independent manner" has now been seen by 4 referees, whose comments are attached. I apologize for the long review process.

After careful consideration and in light of the reviewers' advice we have decided that we cannot offer to publish your manuscript in Nature Genetics.

While the referees find your work of some interest, they raise concerns about the appropriateness of some of the technical approaches, and the strength of the novel conclusions that can be drawn at this stage.

Reviewer #1 thinks this manuscript could potentially bring some clarity to the ongoing PRC2- RNA debate but it would require a major revision. Amongst other things, you would need to test the binding of the EZH2 mutants to different types of RNA, the cell type used (K562) is not ideal (please see below), and you'd need to gain more additional mechanistic insight to fill in some gaps.

We would like to acknowledge Reviewer 1 who stated that our *“work represents an essential piece of the PRC2-RNA puzzle that definitely deserves to be shared with the community”*.

We now added binding assays with additional RNAs (Supplementary Fig. 1e-h) and also carried out experiments using mouse embryonic stem cells, all of which support the original conclusions (see Reviewer 1, point 2 AND the enclosed related manuscript Healy et al.).

Reviewer #2 says the data are of high quality in general, but they think that many key conclusions are not well supported. The reviewer has several experimental suggestions for potential improvement.

We would like to acknowledge Reviewer 2 who stated that *“This study identifies an important feature of EZH2, and PRC2 in general”* and that the *“Quality of the data and reproducibility by the authors are excellent”*. We now included new experiments to address most of the points raised by reviewer 2.

Reviewer #3 feels that the findings are potentially interesting, but that additional controls are required. Like Reviewer #1, they think that the RNA-binding ability of these mutants would need to be tested more in-depth and since Long et al. used pluripotent stem cells, for a fair comparison, you should do the same.

We would like to acknowledge Reviewer 3 who stated that *“Overall this is an interesting study”*. We now carried out all the experiments that were proposed by Reviewer 3, and we believe they improved our manuscript substantially.

Importantly, Reviewer 3 also asked about the ChIP-seq with RNase A treatments (rChIP) that were carried out by Long et al. 2020. We now reproduced these rChIP experiments, including using human iPS cells and mouse ESC. Enclosed is a separate related manuscript draft, describing the rChIP reproduction and our conclusions (Healy et al., enclosed).

Reviewer #4 thinks that some of the key biochemical assays would need to be redone and that the EZH2's RNA-binding properties would need to be more comprehensively characterized. They therefore are skeptical about the current conclusions.

It is clear that Reviewer 4 is a well-established biochemist in the field, and all the concerns they raised are valid. Yet, as we explained in our Appeal Letter, it seems that Reviewer 4 might have

not realised that most of our enzymatic assays were a reproduction of Long et al 2020, and that the technical problems with these assays simply originated from Long et al 2020. We now add some text to better state that. Reviewer 4 might have also missed some of the enzymatic experiments that we carried out under initial rate conditions, as we originally placed them in a supplemental figure (was in Supplementary Fig. 1i, now replaced by data in Fig. 1e). We now carried out all the additional experiments that Reviewer 4 recommended (Fig. 1e), with the new results supporting the original conclusions.

We feel that these reservations are sufficiently important as to preclude publication of this study in Nature Genetics.

I am sorry that we cannot be more positive on this occasion but hope that you will find our referees' comments helpful when preparing your paper for submission elsewhere.

Sincerely, Tiago

Tiago Faial, PhD Chief
Editor Nature Genetics
<https://orcid.org/0000-0003-0864-1200>

While all the specific changes to the manuscript are specified in our response to the reviewers, key changes are listed here:

- **New title:** We now change the title to *“Inseparable RNA binding and chromatin modification activities of a nucleosome-interacting surface in EZH2”*. (The original title was *“An RNA binding surface in EZH2 is required for the deposition of H3K27me3 to chromatin in an RNA independent manner”*.)
- **New Fig 1e:** New data of histone methyltransferase assays under initial rate conditions and three different enzyme concentrations.
- **New Fig 2:** Include data that was merged from Fig 2 and 3 in the old submission.
- **New Fig 4:** Include new ChIP-Rx data (quantitative ChIP-seq with a spike in) for the different EZH2 mutant cell lines using antibodies for SUZ12, EZH2 and H3K27me3.
- **New Fig S1:** New evidence that recombinant PRC2 is devoid of nucleic acid contamination (Fig S1c), new binding assays against various RNAs (Fig S1d-h) and new evidence for the quality of reconstituted chromatin (Fig S1j, k).
- **New Fig S2:** New data of histone methyltransferase assays (Fig S2f), while the rest of the data originated from Fig S1 in the previous version.
- **New Fig S3:** Data that originated from Fig S2 in the previous version.

- New Fig S4:

- > **S4b:** New co-IP experiments for the different mutants.
- > **S4c:** Immunoblotting from clonal-selected cell lines.
- > **S4d:** Characterisation of new EZH2 mutant (mt2*; where only the nucleosome-binding surface is mutated).
- > **S4e:** Knockout with rescue experiments in mouse embryonic stem cells.
- **New Fig S5:** A new Venn diagram in Fig S5e, highlighting the overlap between H3K27me3 peaks within the different EZH2 mutants. The rest of the data is from the old Fig S3.
- **New Fig S6:** New bioinformatics to further characterise the RNA-seq and CUT&Tag (Fig S6b,c), in addition to data from old Fig S4.
- **New Fig S7:** A new figure with new ChIP-Rx data (quantitative ChIP-seq with a spike in) for the different EZH2 mutants using antibodies for SUZ12, EZH2 and H3K27me3.

Reviewers' Comments:

Reviewer #1:

Remarks to the Author:

PRC2 ability to bind RNA has been recently suggested to be essential for PRC2 activity, maintenance of gene expression and proper iPS cells differentiation (Long et al., 2020). In this work, Zhang and colleagues assess in vitro and in vivo activity of two reported RNA binding-defective EZH2 mutants. They provide convincing evidence for the fact that at least one of the two mutations affects PRC2 activity due to impaired chromatin interaction in an RNA-independent manner, thus providing a framework for the reinterpretation of previous results (Long et al., 2020). In its simplicity, the work represents an essential piece of the PRC2-RNA puzzle that definitely deserves to be shared with the community. Yet, several data presented as main figures, could be moved to supplementary figures, and the overall manuscript can be formatted in a 2/3 figures piece.

Nevertheless, this manuscript has two main limitations, which should be addressed experimentally:

1) Binding to RNAs

The ability of EZH2 (wild-type and mutated forms) to bind RNAs is only tested with respect to a single family of RNAs (i.e. G4 24 RNA: UUAGGGx4). The authors should extend this analysis to other types RNAs. It cannot be excluded that point mutations might generate an affinity for other RNA classes.

We now included RNA binding assays using different types of RNA that were previously implicated in binding to PRC2, including regions from HOTAIR and MEG3 lncRNAs that were

proposed to interact with PRC2 (Supplementary Fig. 1e-h):

We also included G4 mt, which is a mutant of G4 RNA that cannot form a G-quadruplex structure. We find that the only RNA that binds PRC2 with high affinity under near-physiological salt concentrations (100 mM KCl) is a G4 RNA, in agreement with recent studies (Wang, Goodrich, et al. 2017; Beltran et al. 2019). However, in the case of HOTAIR lncRNA, where some protein-RNA interactions are detectable, mt1 does exhibit reduced affinity. We should point out that data in the revised manuscript and the previous submission consistently record only a small reduction in the RNA-binding activity for mt2 (generated by Long et al. 2017; 2020). This further strengthens our general conclusion: mt2 dysregulates PRC2 through a defect in a nucleosome interacting surface, rather than perturbed RNA binding.

This is particularly relevant since other PRC2 subunits might retain (or even gain) affinity for RNA. Thus PRC2-RNA binding should be tested *in vivo*, preferably using orthogonal approaches.

We have carried out the fRIP experiments as described in Long et al. 2020, using K562 cells and the same EZH2 antibody used by Long et al. 2020 (Cell Signaling Technologies #5246; data below). Yet, while working on this revision, the lab of Mauro Calabrese published a paper, showing that the same EZH2 antibody cross-reacts with the RNA-binding protein SAFB (Cherney et al. 2023). Using RNA IP with knockout experiments, Cherney et al. also showed that the majority of the RNA that precipitated with this antibody is actually bound to SAFB. Data in Cherney et al. fits with our observations: we observed similar RNA-IP efficiency from all the EZH2 wildtype and mutant cell lines that we tested using the anti-EZH2 antibody that was used in Long et al. 2020 (see figure below). We do not know why the fRIP experiment in Long et al. 2020 identified differences between the EZH2 wildtype to mt2, but we were unable to obtain this observation (we should stress again that Cherney et al. 2023 observed hardly any difference between EZH2 wildtype to EZH2 knockout cells when tested this antibody for RNA-IP). Since the new data by Cherney et al. 2023 imply that the fRIP experiments in Long et al. 2020 are technically flawed, and also since we were

unable to reproduce the fRIP experiments in Long et al. 2020, we wish to leave these results out of the manuscript.

2) K562 bulk cell populations

Analysis of K562 bulk population limit significantly the interpretation of the results. The authors should repeat some of the key experiments using selected clones of K562 or moving to hiPS or mESCs.

We now addressed this concern using the two different approaches that were proposed:

1. Clonal selection in K562 cells: We have selected 4 individual clones for each of the mt1 and mt2 rescued EZH2 knockout K562 lines (Supplementary Fig. 4c, pasted below):

Our K562 clonal selected lines (Supplementary Fig. 4c) phenocopy results obtained in K562 bulk cell population (Fig. 2b and S4a): mt2 leads to the depletion of H3K27me3 while mt1 does not.

2. Knockout with rescue in mESC: We also addressed this concern through knockout with rescue experiments in mESC. Specifically, we rescue the different EZH2 constructs in an *Ezh1* and *Ezh2* double knockout (*Ezh1/2* dKO) mouse embryonic stem cell line. We rescued with either wild type or mutants EZH2. We carried out this experiment twice, using two different lentiviral expression vectors (pHIV-EGFP was the same lentiviral vector that we used for experiments in K562 cells; the pLenti vector provided a higher expression level of EZH2). Importantly, experiments in mESC cells follow the same trend as observed in K562 cells: mt2 phenocopied the catalytically defective dEZH2, while mt1 resembled the EZH2 wild type (Supplementary Fig. 4e; pasted here):

Main points:

1) The authors never analyzed (or mentioned) the effects of mt1 and mt2 mutants on H3K27me1

and H3K27me2.

We now included immunoblotting for H3K27me1 and H3K27me2 (Supplementary Fig. 4a, d):

These results are discussed in line 230.

2) Are the amino-acids mutated in mt1 and mt2, conserved in other species. This might be important to understand when the affinity for nucleosomal DNA/RNAs appear during evolution?

Yes, they are conserved. We have now included the conservation analysis of the amino- acids mutated in mt1 (in blue, below) and mt2 (in red) across species from human to fly (Supplementary Fig. 1a):

This is now discussed in line 101.

3) Which are the levels of EZH1 in K562 cells? Is there any compensation of EZH1 in cells expressing mt1 and mt2 mutants?

Yes, we do see a minor enhancement in the expression level of EZH1 upon knockout of EZH2,

but the expression of EZH1 is diminished upon rescue with EZH2 wild type or mutants. We have now included the immunoblotting of EZH1 in Supplementary Fig. 4a (or see this Figure in the response to Point 1 above). We now discuss this in line 215.

4) The authors should perform cell fractionation to investigate whether wt, mt1, mt2, dEZH2 have different localization/affinity for chromatin.

We have performed cell fractionation and added them into Supplementary Fig. 7a, pasted below:

This analysis does not reveal large changes between the mutants, perhaps with the exception of a small increment of SUZ12 in the soluble fraction in the case of mt2 and dEZH2. This is discussed in line 341.

5) In acute myelogenous leukemia cells, overexpression of PALI1 leads to cell differentiation. The authors should include similar experiments to assess the impact of gene expression changes (upon expression of EZH2 mutants) on the ability of cell to differentiate.

Our new gene ontology (GO) on the RNA-seq data comparing the wild type and the mutants revealed multiple GO terms related to hematopoiesis for mt2 and dEZH2, but not for mt1 (Supplementary Fig. 6b):

The enrichment of hematopoietic-related GO terms for mt2 (e.g. “Blood coagulation” and “Oxygen transport”) fit well with the myelogenous origin of K562 cells and suggests that mt2 pushes them to differentiate. The same analysis on mt1 did not lead to any GO terms as not enough genes were differentially regulated (N=57, see data in Supplementary Fig. S6c), in agreement with mt1 being effective in maintaining cell identity as the wild-type EZH2.

6) As mentioned above, RNA-seq analysis is based on the assumption that PRC2-mediated gene regulation depends on the rescue EZH2. However, Figure 2b and S2d show that the endogenous EZH2 protein is variably expressed. To rule out the possibility that endogenous

EZH2 might be supporting mt1 mutant in aiding PRC2-mediated gene silencing, the authors should repeat the analysis of the transcriptome using selected clones for each genotype.

We have selected 4 individual clones for each mt1 and mt2 rescue lines (see point 2 above). All the clones showed similar EZH2 expression levels, and the selected clones resembled their bulk populations in terms of H3K27me3 levels (Supplementary Fig. 4c). We also carried out similar experiments in mESCs, over the background of *Ezh2* and *Ezh1* knockout, with similar observations: mt1 phenocopied the WT and mt2 phenocopied dEZH2 (Supplementary Fig. 4e). The consistency of observations between the different clones that we selected and across different cell lines reasonably rules out the possibility that residual activity from the endogenous EZH2 substantially supports the activity of mt1 beyond mt2.

7) Provided the data generated for transcriptome analysis, the paper would benefit from a more thorough analysis of the differentially expressed genes other than the sole PCA analysis presented in Figure 4b (e.g. providing numbers and overlap with ChIP-seq peaks of H3K27me3, correlation between loss of H3K27me3 and transcription upregulation, overlap with dEZH2), GO analysis of differentially expressed genes etc... .

We have performed the requested analysis as illustrated in Supplementary Fig. 5e and Supplementary Fig. 6b, c.

In Supplementary Fig. 5e, we showed that in the case of WT and mt1, there are 1235 uniquely overlapping H3K27me3 peaks, whereas only 57 uniquely overlapping H3K27me3 peaks are observed between WT and mt2. There are also 790 uniquely overlapping H3K27me3 peaks between mt2 and dEZH2, while a much lower number of overlapping H3K27me3 peaks between mt1 and dEZH2 (44 peaks). These findings provide further support for mt2 resembling dEZH2, while mt1 resembling WT.

In Supplementary Fig. 6bc, we performed a GO analysis on the differentially expressed genes. The results revealed that both dEZH2 and mt2 lines led to hematopoietic-related GO terms, whereas no significant GO terms were identified for mt1 cells (Supplementary Fig. 6b). This is the result of a low number of genes that were differentially regulated between mt1 to the WT (N=57; Supplementary Fig. 6c), which further supports our conclusion that mt1 phenocopies the WT. Additionally, we showed that genes are more likely to be upregulated than downregulated in the case that the same genes are associated with H3K27me3 in the wild-type cells but lost in the mutant (Supplementary Fig. 6c).

Minor points:

- I would suggest to remove replicates from main figure (e.g. Figure 2b) or merge into single plots where possible (eg. fig. 3b,d)

Done. We have moved all the replicates from Figures 2 and 3 into their corresponding supplementary figures and merged Fig 2 and 3 accordingly.

- Provide a bar-plot quantification to summarize WB data of Figure 2

We deliberately avoided making quantitative arguments based on our Western Blots in Fig. 2 and therefore prefer to avoid this type of quantification.

- Figure 2 and 3 could be easily merged (without replicates) Done.

Figures 2 and 3 have been merged.

- Fig. 4 is not so relevant; it should be moved to supplementary figures.

The data in this Figure (now Fig. 3) indicates that the loss of H3K27me3 in both mt2 and dEZH2 mutant cells leads to changes in the transcription program. We prefer to keep this data in, as it directly supports conclusions made in the manuscript: mt2 phenocopies dEZH2 by defective maintenance of transcription programs.

- Fig. 5a should be reformatted

Done. We have reformatted the model figure.

Typos:

Fig. 1a 'Functiuon' in cells

L226 across all transcription start 'sited' Thank you,

we have corrected this typo.

Reviewer #2:

Remarks to the Author:

A) In the manuscript entitled "An RNA binding surface in EZH2 is required for the deposition of H3K27me3 to chromatin in an RNA-independent manner" Zhang and colleagues try to connect the RNA binding surface of EZH2 to its catalytic activity both in vitro and in vivo. In addition, their findings support the ability of EZH2 (and the PRC2 in general) to bind and modify chromatin in a RNA-independent manner. The main point of this study consists in understanding whether the RNA binding ability of EZH2/PRC2 is involved in targeting the PRC2 to chromatin and that affects its catalytic ability to deposit H3K27 methylation.

B) The major contribution and advancement produced by this study is the "reclassification" of an EZH2 mutant previously considered catalytic active while RNA-binding deficient (termed mt2 in this manuscript and previously characterized in PMID: 32632336) to a catalytic impaired mutant specifically toward nucleosomal incorporated H3. As the authors pointed out, their results disagree with previous findings reported using very similar conditions concerning the activity of the mt2 mutant (PMID: 32632336). This study identifies an important feature of EZH2, and PRC2 in general, that is required for chromatin binding and catalytic activity of the complex. I am less convinced the experiments and results presented ultimately solved the issue of whether PRC2 requires RNA binding to be targeted to chromatin and exert its catalytic function.

We agree about this point: our manuscript cannot, and was not meant to, solve the issue of whether PRC2 required RNA binding to be targeted to chromatin. Instead, we aimed to put the focus here only on a couple of RNA-interacting surfaces in PRC2. Yet, our data implies that the work that was already published by Long et al. 2020 has pushed the field backward:

they made strong conclusions based on a so-called separation-of-function EZH2 mutant that is actually defective in methyltransferase irrespective of RNA (mt2, Fig. 1 in our manuscript). We show here that mt2 disrupts PRC2 function irrespective of its RNA binding activity, which would allow the field to keep moving forward beyond Long et al. 2020. With that in mind, our findings also show that another RNA-binding defective EZH2 mutant, mt1, phenocopied the wild-type EZH2 in cells. This indicates that at least some of the RNA-binding activity of PRC2 can be

reduced without an obvious phenotype in cells.

C) Quality of the data and reproducibility by the authors are excellent. Data presentation is good however including concentrations, gradients of all the reagents in each figure would highly improve and speed up interpretations of the figures by the readers.

We now made changes to figures when applicable, to explicitly state gradients. In other cases, we had to include this information within the legend for graphical considerations.

D) Statistical assessment of the data is appropriate; some figures might require appropriate quantification and/or statistical assessment (Fig. S1B-C)

We included standard errors on the dissociation constants that are presented in Fig. S1d (old Fig. S1b). But we prefer to avoid quantification of the data in Fig. S1i (previously S1c), as we tested there only a limited number of protein concentrations and in some cases the probe was not completely shifted, which would not allow obtaining reliable binding curves.

E) Data appear robust, reliable and valid across figures and panels Thank you.

F) Major points:

1) Based on the data presented by the authors, the best summary of their findings is perfectly represented by the sentence at line 122-125 “This also indicates that the nucleosome-interacting surface of EZH2 is required for the methyltransferase of chromatin, even if this surface is dispensable for the methyltransferase of unchromatinized substrates.” The title and the abstract that refer mainly to the EZH2 RNA binding ability, however the most important data in the manuscript point out to EZH2 and its ability to interact/modify nucleosomes. Both the mutants presented do not massively lose their ability to bind RNA (Fig. S1B, especially mt2 Vs wt). However, the effect of the mt2 EZH2 compared to WT (or to mt1 EZH2) on PRC2 chromatin binding/methyltransferase activity ex vivo (in the cells) and in vitro (nucleosomes) is remarkable. Therefore, it could be concluded that PRC2 containing mt2 EZH2 is defective in chromatin binding despite being quite able to interact with RNA. This could, potentially, be even further demonstrated by measuring the RNA binding ability of the dEZH2 the authors used in this study (there should be no reason why this mutant should not interact with RNA). This supports that EZH2 RNA binding ability is not sufficient for PRC2 targeting to chromatin/methyltransferase activity but does not answer whether it is necessary.

We agree about this point: we think it is very important, and for that reason, we changed the title of the manuscript. We now change the title to “*Inseparable RNA binding and chromatin modification activities of a nucleosome-interacting surface in EZH2*” (the original title was “*An RNA binding surface in EZH2 is required*

for the deposition of H3K27me3 to chromatin in an RNA independent manner”). We also modified the last sentence in the abstract to clarify that phenotype is attributed to a nucleosome-interacting surface. We also added a sentence into the discussion, stating that “our RNA binding assays indicate that EZH2 mt2 is quite able to interact with RNA (Supplementary Fig. 1d,e), despite being defective in chromatin modification.” (Line 378).

As proposed, we now also included the dEZH2 in our RNA binding assays (Supplementary Fig. 1e-g) and indicated in the text that it is “active in RNA binding (Supplementary Fig.1e)” in line 192.

2) Is there any RNA copurified with the PRC2 or the nucleosomes in the in vitro assays? It would be good to include a step with RNase A and/or RNase H in the key in vitro experiments (nucleosome interaction and methyltransferase assays). That would help the authors to understand if there is any RNA contribution to the effect they see in vitro. The main question is can the PRC2 associate to chromatin and modify it without any RNA present in the assay?

To exclude nucleic acid contamination in our PRC2 protein prep, we now added an analysis of the absorbance, including the ratio of absorbance between 260 nm to 280 nm (A_{260}/A_{280} ; Supplementary Fig. 1c):

The low A_{260}/A_{280} ratio (<0.6) and the identical absorbance profiles between PRC2 complexes are indicative of the lack of nucleic acid contamination.

3) An important mutant the authors should consider introducing in the present study is the EZH2 PRKKR494-499NAAIRS without mutations is F32/R34/D36/K39. Based on the cryo-EM structure and the authors' data the interface nucleosomal DNA/EZH2 is very important for PRC2 binding to chromatin and for its catalytic activity. However, based on the cryo-EM structure, only the 6aa PRKKR in EZH2 are part

of the interface. Therefore, including a third mutant (mt3?) could strengthen the authors conclusions, narrow down the effects

described by the authors to a more precise EZH2 protein region and improve the interpretation of the data.

Thank you for this suggestion, we now included the proposed additional mutant, termed mt2*, which is derived based on the sequence of EZH2 mt2 but only contains the EZH2 PRKKKR494-499NAAIRS mutation. We tested mt2* in cells, confirming that mt2* phenocopy mt2 (Supplementary Fig. 4d):

These results suggest that the substrate-nucleosome interacting surface of EZH2 is indeed a key determinant for the PRC2 loss-of-function exhibited by mt2 in cells, and we now discuss it in line 230.

4) The most puzzling results that complicate the interpretation of the study come from the NCP interaction experiments in Fig. S1C. When these results are taken into consideration to interpret all the other results, it is not clear the mechanism behind the effects described by the authors. A simple explanation for the entire study, as mentioned by the authors (Line 122-125), is that they identified a nucleosomal DNA binding region (mediated or not by RNA, answering my question #2 will help to figure this out) of EZH2 that is important for PRC2 binding to chromatin and for its catalytic function. However, both mt1 EZH2 and mt2 EZH2 have the same defective interaction with NCPs (Fig. S1C, I would also invite the authors to quantify these interactions). At this point it would be logical to conclude that the interaction with NCPs is quite dispensable for PRC2 binding to chromatin and for its catalytic activity. This seems unrealistic. However, the conditions used in the EMSA and the methyltransferase assays are different (accordingly to the M&M section).

Can the two assays be performed using comparable concentrations of nucleosome and PRC2?

Is it possible that in the EMSA assays the two different mutants can separately bind nucleic acids (DNA and/or RNA) but while the mt1 EZH2 loses not specific binding ability and therefore not strictly required for PRC2 functions, the mt2 loses important binding ability required for PRC2 interaction and modification of NCPs? The authors should investigate this in more details since it seems to be crucial to their data interpretation/explanation. So far, they just try to explain this important issue in lines 104-106 “This led us to hypothesize that

mt2 might not be able to interact with chromatin in a conformation required for the efficient modification of histone tails”.

It would be interesting competition assays with non-specific DNA/RNA and examine how the two mutants behave in EMSA and HMT assays.

It would be important to use the chromatinized gene (ATOH1) in the EMSA studies.

We agree that our study does not provide an explicit explanation for the somewhat similar nucleosome-interacting profile observed for the different PRC2 constructs that we assayed. However, the interactions between PRC2 to nucleosomes and DNA binding is a rather complex topic, as there are multiple surfaces in PRC2 that interact with nucleosomes and DNA. For instance, Finogenova et al. 2020 have identified regions in the CXC domain of EZH2 that interact with DNA of the substrate nucleosome, and another surface in EED that interact with DNA of the allosteric nucleosome. Poepsel, Kasinath, and Nogales 2018 showed that the N-terminal regions of SUZ12 and RBBP4 interact with nucleosomes.

Consequently, perturbing one nucleosome-interacting surface in PRC2 may not be sufficient to substantially reduce its affinity to nucleosomes, as measured by an EMSA experiment. This was already demonstrated: Finogenova et al. 2020 found that mutating a single nucleosome-interacting surface of PRC2 is sufficient to substantially abolish H3K27me3 methyltransferase in vitro despite only a moderate reduction of the nucleosome binding (less than 3-fold change *Kd* for either the mutants “CXC>A” or “EED>A”; see Fig. 2 in Finogenova et al. 2020).

We agree that further quantitative binding assays under various conditions, including with competitors and different substrates, are required in order to understand how exactly PRC2 interacts with nucleosomes. Yet, this would require substantial development of the study, as EMSA might not be the ideal method for that. For instance, it is very hard to control the salt concentration in an EMSA experiment, as salt tends to diffuse away as soon as the sample is placed in the well, especially when agarose gels are used. Fluorescence anisotropy (FA) does not suffer from this limitation, but it comes with other limitations, such as low signal if the probe is very large. For that reason, a quantitative binding assay on an ATOH1 array would be rather challenging, as the

array may be too large to allow quantifying a band shift in EMSA or anisotropy changes in FA. While there are some solutions for these issues, they would require substantial development and we would prefer to carry them out beyond the scope of this study.

With that being said, our new ChIP-Rx experiments in Fig 4 show that mt2 phenocopy the chromatin occupancy of dEZH2. This suggests that the reduced chromatin occupancy of mt2 in cells is not directly attributed to the defective 'binding' to nucleosomes but it's the other way around: the defective interaction with nucleosomes leads to defective HMTase, which in turn reduces the chromatin occupancy of mt2, as seen for dEZH2 (discussed in a new section that starts in line 324).

5) The rescues experiments in Figure 2 are convincing, however, I would recommend the authors to introduce the mutations by CRISPR especially if the mt3 (see point 3) can recapitulate the effects of mt2. This would be a cleaner system that would avoid co- existence of WT/truncated EZH2 with the EZH2 versions expressed by the rescue constructs.

This is a good suggestion, but both mt1 and mt2 include mutations in amino acids that are coded in multiple exons. Hence introducing these mutations using CRISPR into their endogenous sites would require multiple genome editing cycles. Also Long et al. 2020 did not introduce the mt2 mutation into the endogenous exons, but they rather used CRISPR/Cas to knock in the entire EZH2 open reading frame into exon 2 of the *EZH2* locus. After carrying out the experiments that were suggested above, we now know that mt3 from point 3 (we called it mt2* for simplicity) is indeed an excellent candidate for a CRISPR knockin. Yet, a substantial development would be required to carry out a fair comparison between mt2*, mt1 and mt2 through an endogenous knockin.

G) References are appropriate

H) The manuscript is generally well written, however the constant reference to RNA binding ability and RNA binding surface is quite confusing. Also, the authors might want to reconsider the title that is also not very explanatory. The authors might want to reconsider changing the "RNA binding surface" to "nucleosome binding interface/surface/module" especially if they can further experimentally clarify the points at (F).

Thank you for this suggestion, we have now changed the title to "*Inseparable RNA binding and chromatin modification activities of a nucleosome-interacting surface in EZH2*", which we believe is more informative and will better convey our findings.

Minor comments:

Figure 1A has a typo “functiuon in cells” We

have fixed the typo, thank you.

Reviewer #3:

Remarks to the Author:

In this manuscript, Chen and colleagues addressed the role of RNA-binding in mediating the establishment of H3K27 methylation by PRC2. As the authors stated in the introduction, this is a key and controversial issue, with some recent studies suggesting that RNA-binding is essential for PRC2 recruitment. Here the authors performed biochemical analysis and found that one of the RNA-binding defective mutant EZH2 (mt2) in fact loses the ability to interact with nucleosomal DNA and is unable to methylate H3K27 when mono-nucleosomes and nucleosome arrays are used as substrates. In contrary, another RNA-binding defective mutant EZH2 (mt1), retains the methyltransferase activity. By expressing these mutants in EZH2 knockout K562 cells, authors conclude that EZH2 mt1 but not mt2 is essentially undistinguishable with wildtype EZH2 in terms of restoring levels of H3K27me3 and gene expression, suggesting that RNA-binding activity per se is not required for PRC2 function.

Overall this is an interesting study. I have the following suggestions mainly to enhance the study rigor. While some of the control experiments may seem excessive, given the provocative nature of the conclusion I believe they should be included.

1. The RNA-binding activity of EZH2 mt1 and mt2 needs further assessment. Only one assay was performed and shown in Fig. S1 with somewhat modest reduction in binding. Authors need to include additional assays (eg. EMSA) and test greater diversity of synthetic RNAs (eg. G-quadruplex-forming RNA) to ensure that mt1 is in fact defective in RNA-binding. Even better is to show that EZH2 mt1 mutant binding to RNA is reduced in cells.

We now expanded the RNA binding assays in our work. We carried out the following experiments to address this point:

1. New in vitro RNA-binding assays: We added experiments under different salt concentrations: 200 mM KCl assayed in the original submission (Supplementary Fig. 1d) and new experiments with 100 mM KCl included herein (Supplementary Fig. 1e). We also expanded the panel of RNAs that were assayed (Supplementary Fig. 1e-h), in response to another Reviewer (Reviewer 1, Point 1). All our assays point to a very mild defect in RNA binding by mt2, while mt1 is at least as defective as mt2, if not more. This is on top of the thorough characterisation of these mutants by various binding assays that were already carried out previously, by us on both mt1 and mt2 (Zhang et al. 2019), and by the Cech lab on mt2 (Long et al. 2020 and Long et al. 2017).

2. Attempt reproduction of fRIP experiments from Long et al. 2020: We attempted to reproduce the RNA-

IP with formaldehyde crosslinking (fRIP) from cells as carried out by Long et al. 2020, using the same EZH2 antibody. Yet, we did not identify any difference in the RNA-IP between any of the mutants (inc. mt2), which is in agreement with a new paper that identified a strong cross-reactivity between this EZH2 antibody to the RNA-binding protein SAFB (Cherney et al. 2023). Specifically, using RNA-IP with EZH2 knockout and SAFB knockout experiments, Cherney et al demonstrated that the majority of RNA that is IP by this antibody is actually bound to SAFB, not EZH2. We do not know why Long et al observed different fRIP efficiency between EZH2 mt2 to WT despite this antibody predominantly reacting with SAFB. We were unable to reproduce their results (see data from our reproduction in the response to Reviewer 1, point 1).

3. Reproduction of rChIP experiments from Long et al 2020: Long et al. 2020 also carried out ChIP in the presence and absence of RNase A treatment (they called it “rChIP”). Using this rChIP experiment, Long et al. 2020 identified the depletion of EZH2 and SUZ12 ChIP signals after RNase A treatment. We are enclosing herein another related manuscript from our lab (Healy et al), showing that the depletion of EZH2 and SUZ12 ChIP-signals after RNase A treatment is an artefact of chromatin precipitation. More information about the rChIP reproduction is provided below, in response to point 6.

2. Quality control data is missing for the in vitro reconstituted chromatin using ATOH1 locus. What is the density and distribution patterns of nucleosomes on this synthetic DNA template?

We now added analytical ultracentrifugation (AUC) and MNase digestion of the ATOH1 array (Supplementary Fig. 1j-k). These results demonstrate a quality reconstitution of the ATOH1 array with nucleosomes that are not regularly phased, in agreement with previous studies that characterised nucleosome arrays that were reconstituted using native DNA sequences (Krietenstein et al. 2016).

3. Given that PRC2 activity can be allosterically activated by H3K27me3, could authors test the effect of spiking in H3K27me3 peptides or nucleosomes on the methyltransferase activity of mt1 and mt2 EZH2?

We have now included an *in vitro* histone methyltransferase assay using a chromatinized ATOH1 DNA as the substrate, in the presence or absence of the allosteric effector peptide H3K27me3 (Supplementary Fig. 2f). As expected, mt2 is substantially less active than the wildtype in the presence of the allosteric peptide. We also noticed an elevated activity of mt1 with respect to the wild type, possibly owing to the location of this mutation in the vicinity of the allosteric regulatory centre of PRC2 (Zhang et al. 2019).

4. In the publication by Long et al. 2020 which is heavily referred to in this manuscript, human pluripotent stem cells were used for cellular functional studies. Here authors used K562 cells. For a fair comparison, could the authors study mt1 and mt2 EZH2 in a stem cell context?

Yes, we now added experiments using mouse embryonic stem cells (Supplementary Fig. 4), which are in agreement with our observations in K562 cells. For complete details, please see our response to Reviewer 1, Point 2.

5. In CUT&Tag studies, only parental K562 was included as control. K562 cells expressing EZH2 gRNA should also be included as additional control.

We have now included enrichment profiles for the K562 cells expressing EZH2 gRNA without EZH2 rescue (Supplementary Fig. 5c; grey line). Stating the obvious: EZH1 is upregulated upon the knockout (KO) of EZH2, if no EZH2 rescue is carried out simultaneously (see Supplementary Fig. 4a). This could explain why H3K27me3 is lower in the EZH2 KO with dEZH2 rescue with respect to the EZH2 KO without rescue (compare the grey to the orange lines in the new Supplementary Fig. 5c, below). For the same reason, a fair comparison should be made between

the different EZH2 rescues (Fig. 2c-e). The text has been added in line 259 to reflect this.

6. In addition to H3K27me3, authors should perform (1) co-IP to test the PRC2 complex assembly with EZH2 mt1 and mt2 in cells and (2) ChIP to examine genome-wide binding patterns of EZH2 mt1 and mt2. These studies would be quite informative. Related, in Long et al. 2020 study, ChIP for PRC2 was performed with and without treatment of RNase, I wonder if RNase treatment would affect the recruitment of EZH2 mt1 mutant?

We now carried out all these experiments accordingly:

1. Co-IP for PRC2 subunits: We have performed co-IP for checking whether the EZH2 mutants affect the assembly of PRC2 complexes in cells (Supplementary Fig. 4b). As expected, all the EZH2 constructs can form a complex with SUZ12 (PRC2) but not CBX7 (PRC1). We mentioned that in the text, in line 209.

2. ChIP-seq: We performed quantitative ChIP-seq with exogenous reference genome spike-in (ChIP-Rx) to analyse the genome-wide binding of PRC2 (EZH2 and SUZ12; Fig. 4 and Supplementary Fig. 7). As a control, we also did ChIP-Rx for H3K27me3. The new ChIP-Rx (Fig. 4) is in agreement with our CUT&Tag data (Fig. 2): mt2 resembles the catalytic defective dEZH2 mutant while mt1 resembles the wild-type EZH2. Note that the chromatin occupancy of PRC2 follows the same trend, where dEZH2 phenocopy mt2. The reduced chromatin occupancy of both dEZH2 and mt2 suggests that this phenomenon is more likely attributed to defective HMTase rather than to defective RNA binding. We have added a whole new section, starting in line 324, to describe these results.

3. ChIP-seq with RNase A treatment (rChIP): We now reproduced the rChIP experiment that was carried out by Long et al 2020. Enclosed is another manuscript from our lab (Healy et al.), titled “*Apparent loss of PRC2 chromatin occupancy as an artefact of RNA depletion*”. In Healy et al., we describe an extensive reproduction of the ChIP-seq experiments carried out by Long et al. 2020, including the RNase A treatment they carried out (termed “rChIP” in Long et al. 2020). Briefly, we reproduced that rChIP experiment in **mouse embryonic stem cells** (mESC; Fig. 2 at Healy et al), **human iPS cells** (Fig. 3 and 4 at Healy et al) and human K652 cells (Fig. 3 in Healy et al). We found that the depletion of PRC2 ChIP-signals after RNase A treatment is the result of an experimental artefact: RNase A treatment under the experimental conditions used by Long et al led to the precipitation of non-targeted chromatin on the beads. In turn, during normalisation, the massive gain of total genomic DNA leads to the reduction of all ChIP signals from facultative heterochromatin (Fig. 4 in Healy et al). Importantly, we included in all our rChIP experiments an H3K27me3 control (Fig. 2-4; Long et al did not include that control). The H3K27me3 rChIP control demonstrates that RNase treatment leads to the depletion of H3K27me3 as a result of the same artefact (from Fig. 2 in Healy et al):

A

In Healy et al. we also demonstrate that under other ChIP-seq conditions, where more salt is present during the sonication, RNA depletion using RNase A does not affect the PRC2 chromatin occupancy as identified by ChIP-seq (Fig. 1 in Healy et al). Collectively, the enclosed manuscript (Healy et al.) supports two key conclusions: (i) RNA is dispensable for the PRC2 chromatin occupancy as captured during ChIP-seq experiment (Fig. 1 in Healy et al); and (ii) Fig 1 in Long et al. 2020 has been established based on an experimental artefact. This is important as Fig 1 in Long et al. 2020 supported part of the title of that paper: “*RNA is essential for PRC2 chromatin occupancy ...*”. Because reproducing that rChIP included much work and somewhat different author contributions, we would like to publish Healy et al. separately.

7. In previous publications by the author’s group, it was proposed that promiscuous RNA binding by PRC2 represents a mechanism to exclude PRC2 recruitment to transcriptionally active regions. According to this model, EZH2 mt1 would be predicted to localize to at least some active genes? Can authors analyze the H3K27me3 data to see if this is true?

Thank you for this suggestion, we now carried out this analysis (Fig. 4c). We first performed quantitative ChIP-seq with exogenous reference genome spike-in (ChIP-Rx) for H3K27me3, SUZ12 and EZH2 (Fig. 4 and Supplementary Fig. 7). Next, we split all transcription start sites (TSS) in the genome into three groups, based on H3K27ac level as a proxy for transcription: top 25% (highly expressed genes), mid 50 % (moderately expressed genes) and bottom 25% (lowly expressed genes). As expected, this analysis (in Fig. 4c) revealed that PRC2 has increased

chromatin occupancy at lowly expressed genes. Yet, in all TSS groups, we observed the same trend for the different EZH2 constructs: EZH2 mt1 closely resembled the WT while mt2 resembled dEZH2 (we add a new section in lines 337 to discuss these results):

Reviewer #4:

Remarks to the Author:

Main concerns:

1. How many RNA-binding domains are there within EZH2? How do the authors know that they studied the relevant RNA-binding domain?

EZH2 and other subunits of PRC2 contain multiple RNA-binding surfaces (e.g. Kaneko et al. 2010; 2014; Long et al. 2017; Zhang et al. 2019; Kasinath et al. 2018; Song et al. 2023). In this work, we placed most of the focus on an RNA binding site that is represented by the EZH2 mutants mt2 or mt2* here. The focus on this site/mutant was made for two reasons: (i) Long et al 2020 proposed that mt2 is a separation-of-function mutant, active in methyltransferase but defective in RNA

binding. (ii) Mutated residues of mt2 reside within a nucleosome binding site of EZH2, while interplay and competition between RNA to nucleosomes were previously proposed in the case of PRC2 (e.g. Beltran et al. 2016; 2019; Wang, Paucek, et al. 2017). We reasoned that mt2 may be an important mutant for the mechanistic study of PRC2, given its previously unappreciated role in HMTase through interactions with the substrate nucleosome (Fig. 1b).

We are aware of mutations in other RNA-binding sites of PRC2, which were identified by the Reinberg lab, as an RNA-binding defective EZH2 (Kaneko et al. 2010) and JARID2 (Kaneko et al. 2014), but these are beyond the scope of this study. To better state the focus

of the study, we now changed the title of the manuscript to: *“Inseparable RNA binding and chromatin modification activities of a nucleosome-interacting surface in EZH2”* (the original title was *“An RNA binding surface in EZH2 is required for the deposition of H3K27me3 to chromatin in an RNA independent manner”*).

2. There are different assays used utilizing different substrates, octamers (or H3), which is irrelevant as PRC2 is not active with core histones or octamers, mononucleosomes and oligonucleosomes. The conditions for each assay are different, but most importantly, when analyzing enzymes and comparing wild-type to mutants, the assays must be performed during the initial part of the reaction (rate), not incubated for 2 hrs (yield in real biochemical terms).

We might not have clarified this well enough in the original submission, but all the mentioned experiments were a reproduction of experiments that were carried out by (Long et al. 2020). We now added text to better state that (lines 116 and 152).

We agree about the requirement to carry out enzymatic experiments under initial rate conditions, and for that reason, we actually included such assays in the original submission (old Supplementary Fig. 1i). We might have made a poor choice to place that analysis in a supplementary figure, so we now extended this analysis and placed it as a body figure; thank you for pointing this out (see Fig. 1e):

Note that the original conclusions still hold: mt2 is substantially less active than the wild-type PRC2 on oligo-nucleosome substrates.

3. The most surprising aspect to me is that the authors appear to forget a basic principle of biochemistry, when comparing different mutants, as is in this study, one must use different concentrations of proteins and compare their activity, and the assay must be performed under rate conditions. None of these basic concepts of biochemistry were applied in this study. Thus, I do not believe the conclusions.

We again wish to reiterate that most of the enzymatic experiments were a reproduction of Long et al. 2020, while some additional HMTase assays were carried out under initial rate conditions (original Supplementary Fig. 1i). Nevertheless, we now added more experiments under initial rate conditions AND using different enzyme concentrations (Fig. 1e and above). This new data supports the original conclusions: mt2 is substantially less active than the wild-type PRC2 on an oligo-nucleosomal substrate. The mt1 mutant resembles the wild-type PRC2.

References

- Beltran, Manuel, Manuel Tavares, Neil Justin, Garima Khandelwal, John Ambrose, Benjamin M. Foster, Kaylee B. Worlock, et al. 2019. 'G-Tract RNA Removes Polycomb Repressive Complex 2 from Genes'. *Nature Structural & Molecular Biology* 26 (10): 899–909. <https://doi.org/10.1038/s41594-019-0293-z>.
- Beltran, Manuel, Christopher M. Yates, Lenka Skalska, Marcus Dawson, Filipa P. Reis, Keijo Viiri, Cynthia L. Fisher, et al. 2016. 'The Interaction of PRC2 with RNA or Chromatin Is Mutually Antagonistic'. *Genome Research* 26 (7): 896–907. <https://doi.org/10.1101/gr.197632.115>.
- Cherney, Rachel E., Christine A. Mills, Laura E. Herring, Aki K. Bracerros, and J. Mauro Calabrese. 2023. 'A Monoclonal Antibody Raised against Human EZH2 Cross- Reacts with the RNA-Binding Protein SAFB'. *Biology Open* 12 (6): bio059955. <https://doi.org/10.1242/bio.059955>.
- Finogenova, Ksenia, Jacques Bonnet, Simon Poepsel, Ingmar B. Schäfer, Katja Finkl, Katharina Schmid, Claudia Litz, Mike Strauss, Christian Benda, and Jürg Müller. 2020. 'Structural Basis for PRC2 Decoding of Active Histone Methylation Marks H3K36me2/3'. *ELife* 9 (November): e61964. <https://doi.org/10.7554/eLife.61964>.
- Kaneko, Syuzo, Roberto Bonasio, Ricardo Saldaña-Meyer, Takahaki Yoshida, Jinsook Son, Koichiro Nishino, Akihiro Umezawa, and Danny Reinberg. 2014. 'Interactions between JARID2 and Noncoding RNAs Regulate PRC2 Recruitment to Chromatin'. *Molecular Cell* 53 (2): 290–300. <https://doi.org/10.1016/j.molcel.2013.11.012>.
- Kaneko, Syuzo, Gang Li, Jinsook Son, Chong-Feng Xu, Raphael Margueron, Thomas A. Neubert, and Danny Reinberg. 2010. 'Phosphorylation of the PRC2 Component Ezh2 Is Cell Cycle-Regulated and up-Regulates Its Binding to NcRNA'. *Genes & Development* 24 (23): 2615–20. <https://doi.org/10.1101/gad.1983810>.
- Kasinath, Vignesh, Marco Faini, Simon Poepsel, Dvir Reif, Xinyu Ashlee Feng, Goran Stjepanovic, Ruedi Aebersold, and Eva Nogales. 2018. 'Structures of Human PRC2 with Its Cofactors AEBP2 and JARID2'. *Science (New York, N. Y.)* 359 (6378): 940–44. <https://doi.org/10.1126/science.aar5700>.
- Krietenstein, Nils, Megha Wal, Shinya Watanabe, Bongsoo Park, Craig L. Peterson, B. Franklin Pugh, and Philipp Korber. 2016. 'Genomic Nucleosome Organization Reconstituted with Pure Proteins'. *Cell* 167 (3): 709–721.e12. <https://doi.org/10.1016/j.cell.2016.09.045>.

- Long, Yicheng, Ben Bolanos, Lihu Gong, Wei Liu, Karen J. Goodrich, Xin Yang, Siming Chen, et al. 2017. 'Conserved RNA-Binding Specificity of Polycomb Repressive Complex 2 Is Achieved by Dispersed Amino Acid Patches in EZH2'. *ELife* 6 (November): e31558. <https://doi.org/10.7554/eLife.31558>.
- Long, Yicheng, Taeyoung Hwang, Anne R. Gooding, Karen J. Goodrich, John L. Rinn, and Thomas R. Cech. 2020. 'RNA Is Essential for PRC2 Chromatin Occupancy and Function in Human Pluripotent Stem Cells'. *Nature Genetics* 52 (9): 931–38. <https://doi.org/10.1038/s41588-020-0662-x>.
- Poepsel, Simon, Vignesh Kasinath, and Eva Nogales. 2018. 'Cryo-EM Structures of PRC2 Simultaneously Engaged with Two Functionally Distinct Nucleosomes'. *Nature Structural & Molecular Biology* 25 (2): 154–62. <https://doi.org/10.1038/s41594-018-0023-y>.
- Song, Jiarui, Anne R. Gooding, Wayne O. Hemphill, Vignesh Kasinath, and Thomas R. Cech. 2023. 'Structural Basis for Inactivation of PRC2 by G-Quadruplex RNA'. *bioRxiv*. <https://doi.org/10.1101/2023.02.06.527314>.
- Wang, Xueyin, Karen J. Goodrich, Anne R. Gooding, Haroon Naeem, Stuart Archer, Richard D. Paucek, Daniel T. Youmans, Thomas R. Cech, and Chen Davidovich. 2017. 'Targeting of Polycomb Repressive Complex 2 to RNA by Short Repeats of Consecutive Guanines'. *Molecular Cell* 65 (6): 1056-1067.e5. <https://doi.org/10.1016/j.molcel.2017.02.003>.
- Wang, Xueyin, Richard D. Paucek, Anne R. Gooding, Zachary Z. Brown, Eva J. Ge, Tom W. Muir, and Thomas R. Cech. 2017. 'Molecular Analysis of PRC2 Recruitment to DNA in Chromatin and Its Inhibition by RNA'. *Nature Structural & Molecular Biology* 24 (12): 1028–38. <https://doi.org/10.1038/nsmb.3487>.
- Zhang, Qi, Nicholas J. McKenzie, Robert Warneford-Thomson, Emma H. Gail, Sarena F. Flanigan, Brady M. Owen, Richard Lauman, et al. 2019. 'RNA Exploits an Exposed Regulatory Site to Inhibit the Enzymatic Activity of PRC2'. *Nature Structural & Molecular Biology* 26 (3): 237–47. <https://doi.org/10.1038/s41594-019-0197-y>.

Decision Letter, first revision:

1st Nov 2023

Dear Chen,

Thank you for submitting your revised manuscript entitled "Inseparable RNA binding and chromatin modification activities of a nucleosome-interacting surface in EZH2" (NG-A60347R1). It has now been seen by the original referees and their comments are below. The reviewers find that the paper has improved in revision, and therefore we'll be happy in principle to publish it in *Nature Genetics*, pending minor revisions to satisfy the referees' final requests and to comply with our editorial and formatting guidelines.

Importantly, reviewer #2 has one final request/suggestion, which is to either remove Fig. 4c or add a control (EZH2 KO cells). This needs to be carefully addressed.

Since the current version of your manuscript is in a PDF format, please email us (natgen@us.nature.com) a copy of the file in an editable format (Microsoft Word)-- we can not proceed with PDFs at this stage.

We will then be performing detailed checks on your paper and will send you a checklist detailing our editorial and formatting requirements soon. Please do not upload the final materials and make any revisions until you receive this additional information from us.

Thank you again for your interest in Nature Genetics. Please do not hesitate to contact me if you have any questions.

Congratulations!

Sincerely,

Tiago

Tiago Faial, PhD
Chief Editor
Nature Genetics
<https://orcid.org/0000-0003-0864-1200>

Reviewer #1 (Remarks to the Author):

The authors have done a good job in addressing our criticisms.

Reviewer #2 (Remarks to the Author):

The authors did a great job addressing all the reviewers' comments and concerns. The study, as I already mentioned, is solid and addresses an important issue in the field.

I have only one important point that, even though it is not central to this study, I believe the authors should address before publication.

In Figure 4C there is some PRC2 (mainly EZH2 but also SUZ12) occupancy at expressed gene promoters (H3K27ac positive) especially at the top 25%.

I am aware that there are studies showing that EZH2 can bind "solo" at promoters of some transcriptionally active genes (PMID: 35210568; 23239736). However, I would urge the authors to include relevant controls (i.e. *EZH2* KO cells that the authors already generated) to determine the background of the ChIP-seq experiments. The fact that the average EZH2 occupancy in WT cells remains similar independently of H3K27ac occupancy is quite surprising.

Reviewer #3 (Remarks to the Author):

In the revised manuscript, the authors performed an impressive amount of additional work to strengthen the rigor of the study. I have no more comments/suggestions on the technical aspects. The main conclusion, as authors pointed out, is that an EZH2 mutation previously believed as defective in RNA-binding but methyltransferase competent, is actually methyltransferase dead and only slightly reduced in RNA-binding activity. Since none of the mutants characterized in this study completely abolishes RNA-binding yet retains methyltransferase activity, the work does not directly address if RNA-binding is required for PRC2 activity (as reflected by the title change). For this reason and the recent work by Cherney et al. 2023 and Healy et al. 2023 (also from the Davidovich group), I do want to note that the impact and scope of the work are more limited following the revision.

Author Rebuttal, first revision:

We would like to thank all the reviewers for dedicating their time to review our revised manuscript and for their positive view of the additional work that we did during the revision.

Reviewer #1:

Remarks to the Author:

The authors have done a good job in addressing our criticisms. Thank you.

Reviewer #2:

Remarks to the Author:

The authors did a great job addressing all the reviewers' comments and concerns. The study, as I already mentioned, is solid and addresses an important issue in the field.

Thank you.

I have only one important point that, even though it is not central to this study, I believe the authors should address before publication.

In Figure 4C there is some PRC2 (mainly EZH2 but also SUZ12) occupancy at expressed gene promoters (H3K27ac positive) especially at the top 25%.

I am aware that there are studies showing that EZH2 can bind "solo" at promoters of some transcriptionally active genes (PMID: 35210568; 23239736). However, I would urge the authors to include relevant controls (i.e. *EZH2* KO cells that the authors already generated) to determine the background of the ChIP-seq experiments. The fact that the average EZH2 occupancy in WT cells remains similar independently of H3K27ac occupancy is quite surprising.

We now removed Fig 4c and the text that referred to it. For the same reason, we also removed Fig S7b, which was referred to within the legend of Fig 4c and presented representative genomic tracks that were classified based on Fig 4c.

Reviewer #3:

Remarks to the Author:

In the revised manuscript, the authors performed an impressive amount of additional work to strengthen the rigor of the study. I have no more comments/suggestions on the technical aspects. The main conclusion, as authors pointed out, is that an EZH2 mutation previously believed as defective in RNA-binding but methyltransferase competent, is actually methyltransferase dead and only slightly reduced in RNA-binding activity. Since none of the mutants characterized in this study completely abolishes RNA-binding yet retains methyltransferase activity, the work does not directly address if RNA-binding is required for PRC2 activity (as reflected by the title change). For this reason and the recent work by Cherney et al. 2023 and Healy et al. 2023 (also from Davidovich group), I do want to note that the impact and scope of the work are more limited following the revision.

Thank you. We acknowledge that future works would be required for determining how, and to what extent, PRC2 is regulated by RNA. In that sense, we believe that our work will significantly impact the field by enabling it to move forward beyond unsubstantiated paradigms.

Final Decision Letter:

2nd April, 2024

Dear Dr. Davidovich,

I am delighted to say that your manuscript "Inseparable RNA binding and chromatin modification activities of a nucleosome-interacting surface in EZH2" has been accepted for publication in an upcoming issue of Nature Genetics.

Your paper will be published online after we receive your corrections and will appear in print in the next available issue. You can find out your date of online publication by contacting the Nature Press

Office (press@nature.com) after sending your e-proof corrections.

Please note that *Nature Genetics* is a Transformative Journal (TJ). Authors may publish their research with us through the traditional subscription access route or make their paper immediately open access through payment of an article-processing charge (APC). Authors will not be required to make a final decision about access to their article until it has been accepted. Find out more about Transformative Journals

Authors may need to take specific actions to achieve compliance with funder and institutional open access mandates. If your research is supported by a funder that requires immediate open access (e.g. according to Plan S principles) then you should select the gold OA route, and we will direct you to the compliant route where possible. For authors selecting the subscription publication route, the journal's standard licensing terms will need to be accepted, including [a href="https://www.nature.com/nature-portfolio/editorial-policies/self-archiving-and-license-to-publish](https://www.nature.com/nature-portfolio/editorial-policies/self-archiving-and-license-to-publish). Those licensing terms will supersede any other terms that the author or any third party may assert apply to any version of the manuscript.

If you have not already done so, we invite you to upload the step-by-step protocols used in this manuscript to the Protocols Exchange, part of our on-line web resource, natureprotocols.com. If you complete the upload by the time you receive your manuscript proofs, we can insert links in your article that lead directly to the protocol details. Your protocol will be made freely available upon publication of your paper. By participating in natureprotocols.com, you are enabling researchers to more readily reproduce or adapt the methodology you use. [Natureprotocols.com](http://natureprotocols.com) is fully searchable, providing your protocols and paper with increased utility and visibility. Please submit your protocol to <https://protocolexchange.researchsquare.com/>. After entering your nature.com username and password you will need to enter your manuscript number (NG-A60380R3). Further information can be found at <https://www.nature.com/nature-portfolio/editorial-policies/reporting-standards#protocols>

Sincerely,

Tiago

Tiago Faial, PhD

Chief Editor

Nature Genetics

<https://orcid.org/0000-0003-0864-1200>